# Homogeneously catalysed conversion of aqueous formaldehyde to $H_2$ and carbonate

M. Trincado[1], Vivek Sinha[2], Rafael E. Rodriguez-Lugo[3], Bruno Pribanic[1], Bas de Bruin[2] & Hansjörg Grützmacher[1]

Small organic molecules provide a promising solution for the requirement to store large amounts of hydrogen in a future hydrogen-based energy system. Herein, we report that diolefin–ruthenium complexes containing the chemically and redox non-innocent ligand $trop_2dad$ catalyse the production of $H_2$ from formaldehyde and water in the presence of a base. The process involves the catalytic conversion to carbonate salt using aqueous solutions and is the fastest reported for acceptorless formalin dehydrogenation to date. A mechanism supported by density functional theory calculations postulates protonation of a ruthenium hydride to form a low-valent active species, the reversible uptake of dihydrogen by the ligand and active participation of both the ligand and the metal in substrate activation and dihydrogen bond formation.

[1] Department of Chemistry and Applied Biosciences ETH Zürich, Laboratory of Inorganic Chemistry, Wolfgang Pauli Str. 10, Zürich CH-8093, Switzerland. [2] Supramolecular and Homogeneous Catalysis Group, van 't Hoff Institute for Molecular Sciences (HIMS), University of Amsterdam, Science Park 904, Amsterdam 1098 XH, The Netherlands. [3] Labotatorio de Química Bioinorgánica, Centro de Química, Instituto Venezolano de Investigaciones Científicas (IVIC), Caracas 1020-A, Venezuela. Correspondence and requests for materials should be addressed to M.T. (email: trincado@inorg.chem.ethz.ch) or to B.d.B. (email: B.deBruin@uva.nl) or to H.G. (email: hgruetzmacher@ethz.ch).

The use of water as oxygen transfer reagent is a highly promising approach to develop oxygenation reactions of organic substrates under mild and environmentally benign conditions[1]. The investigation of such methods may also give fundamental insight into the splitting of water into $O_2$ and $H_2$ (refs 2,3). While steam reforming that converts hydrocarbons and water into oxygenated products and hydrogen or the water gas shift reaction over heterogeneous catalysts is well developed[4], limited progress has thus far been made with homogeneous catalysts[5]. These may not only operate under much milder conditions but usually also give easier access to valuable mechanistic information[6–8].

The conversion of methanol/water mixtures into $CO_2$ and hydrogen is an important process with respect to the use of liquid organic fuels (LOFs) according to LOF-$H_2$ $\rightleftarrows$ LOF + $H_2$ (LOF-$H_2$ = fuel; LOF = spent fuel). This reaction can be promoted under mild conditions ($<100\,°C$) by well-defined group 8 or 9 metal complexes $\mathbf{A}$[9–12], $\mathbf{B}$[13,14] $\mathbf{1K}$[15] and $\mathbf{D}$[16] bearing a functional cooperative ligand (Fig. 1). Catalyst $\mathbf{A}$ (M = Ru) and analogues catalyse also the back-reaction, that is, the hydrogenation of $CO_2$ to methanol with turnover (TON) of > 2000 (refs 17,18). [Co(triphos)(OOCR)](BF$_4$) also catalyses the hydrogenation of carboxylic acids to alcohols and water, including formic acid (TON > 200)[19]. The complex $\mathbf{1K}$, with a redox and chemically non-innocent diolefin-diazadiene ligand, trop$_2$dad = 1,4-bis(5$H$-dibenzo[a,d]cyclohepten-5-yl)-1,4-diazabuta-1,3-diene[20], converts selectively MeOH/water mixtures into hydrogen and $CO_2$. Of the few homogeneous systems known, $\mathbf{1K}$ does not contain a phosphane as ligand[21] and is the only example that catalyses this reaction in the absence of any additives.

Apart from methanol, formaldehyde is an interesting hydrogen storage molecule (a 1:1 formaldehyde/water mixture contains 8.3 wt% of hydrogen). The dihydrogen-releasing reaction with water ($H_2CO + H_2O \rightleftarrows CO_2 + 2H_2$) is strongly exothermic ($\Delta H_r = -35.8\,kJ\,mol^{-1}$), thus providing a strong driving force for $H_2$ production. This contrasts with dihydrogen release from methanol/water mixtures, which is endothermic ($\Delta H_r = +53.3\,kJ\,mol^{-1}$) and thus requires harsher conditions (Supplementary Note 1). Formaldehyde has been intensively studied due to its importance in the atmosphere, interstellar space and combustion chemistry[22] and plays a key role in the metabolism of living cells[23], acting as a source of reduction equivalents and C1 carbon feedstock for carbohydrates[24]. Few reports describe the conversion of aqueous aldehyde solutions to hydrogen and oxygenated products. Maitlis and coworkers have reported the conversion of acetaldehyde to acetic acid promoted by half-sandwich complexes of Ru[25]. More recently, Prechtl et al.[26] described the dehydrogenation of aqueous methanol to formaldehyde under the release of $H_2$ under mild conditions (298–368 K). In this process, an oxidase enzyme is combined with the $p$-cymene Ru complex $\mathbf{C}$ to achieve the transformation of methanol to formaldehyde hydrate (enzyme catalysed), which can be further converted to $CO_2$ at room temperature in a process catalysed by $\mathbf{C}$ using a hydrogen acceptor. To date, complex $\mathbf{C}$ is the only catalyst that promotes the dehydrogenation of aqueous formaldehyde solutions to $CO_2$ and $H_2$ with acceptable TON and turnover frequencies (TOFs) at 95 °C (pH = 5.5)[27,28]. With more diluted aqueous formaldehyde solutions (1.6 M) TONs of 700 and TOFs of 3142 h$^{-1}$ could be achieved even without additives[29]. The reaction of formalin to $CO_2$/$H_2$ is also catalysed by complex $\mathbf{D}$ under similar reaction conditions and with acceptable TONs (178) but with very low reaction rates. The water soluble iridium(III) hydroxo complex ($\mathbf{E}$) is able to produce $H_2$ and $CO_2$ in a 2:1 ratio from paraformaldehyde. Although this reaction is associated with very low TONs (up to 24), it proceeds at room

temperature and the rate of $H_2$ production increases with increasing pH[30]. In the reactions with $\mathbf{C}$ and $\mathbf{E}$, metal hydrides were detected that led to the proposition of a classical mechanism in which the substrate is activated and converted at the metal centre. By contrast, the bipyridonate in $\mathbf{D}$ is proposed to act as a cooperative ligand, fulfilling the role of an internal Brønsted base/acid in key steps of the catalytic reaction (see Fig. 1).

Metal complexes with cooperative ligands are the active sites in many enzymatic reactions. The mechanisms underpinning these remarkable and new transformations (reactions (a) and (b) in Fig. 1) are still under debate (Fig. 1). But there is an increasing evidence and consensus that metal–ligand cooperativity is involved in the key steps of these reactions[31–33]. A simplified representation of metal–ligand cooperativity is shown in Fig. 1c. The cooperative site in the ligand can be adjacent to the metal centre as the amide function in complex $\mathbf{A}$[34] or slightly remote as suggested for complex $\mathbf{B}$. Previous calculations imply that catalysis with complex $\mathbf{1K}$ may be a special case where the conversion of the substrate is mediated by the ligand exclusively[35,36]. In every case, the substrate binds in the second coordination sphere of the catalytically active complex, becomes dehydrogenated and is released. With methanol as substrate, first formaldehyde is formed in a dehydrogenation reaction, which is subsequently hydrated to the geminal diol. The latter is dehydrogenated to give formic acid, which decomposes to $CO_2$ and $H_2$ in an exergonic catalytic follow-up reaction. Formally, water serves as the oxygen donor in the overall alcohol oxygenation to $CO_2$, with release of $H_2$ as the desired product (Fig. 1c)[37].

Herein we report a new catalytic system, which allows the formaldehyde–water shift reaction under mild conditions with good-to-excellent yields and high TOFs. A mechanism based on experimental observations and density functional theory (DFT) calculations is proposed to explain the dehydrogenation of methanediol to formate and finally $CO_2$ in a basic aqueous solution.

## Results

**Stoichiometric experiments.** The previously reported and easily accessible complex $\mathbf{1K}$, which is active as dehydrogenation catalyst of methanol/water mixtures, forms a tight ion pair between the [Ru(H)(trop$_2$dad)]$^-$ anion and the [K(dme)$_2$]$^+$ cation. The structural data clearly show that the trop$_2$dad ligand coordinates in its reduced enediamide form (trop-N$^-$-CH = CH–N$^-$-trop) to the Ru(II) centre [trop$_2$dad = (1,4-bis(5$H$-dibenzo[a,d]cyclohepten-5-yl)-1,4-diazabuta-1,3-diene][15]. This complex serves as starting material for a number of new Ru(II)-trop$_2$dad derivatives as shown in Fig. 2.

In order to obtain ion separated complexes that contain the [RuH(trop$_2$dad)]$^-$ anion in non-coordinated form, $\mathbf{1K}$ was reacted with dibenzo-18-crown-6 (db18-C-6) or quaternary ammonium salts, [R$_4$N]Br, to give the deep orange $\mathbf{1KC}$ or burgundy red complexes $\mathbf{1Aa}$ and $\mathbf{1Ab}$ as shown in Fig. 2, where $\mathbf{Aa}$ and $\mathbf{Ab}$ indicate the quaternary ammonium cation $n$Bu$_4$N$^+$ or Me$_3$(dodecyl)N$^+$, respectively. These compounds show slightly but significantly more deshielded $^1$H NMR signals for the hydride ligand, Ru-H ($\mathbf{1KC}$: $\delta = -9.00$ p.p.m.; $\mathbf{1Aa}$: $\delta = -9.37$ p.p.m.; $\mathbf{1Ab}$: $\delta = -9.65$ p.p.m.) when compared to $\mathbf{1K}$ ($\delta = -10.25$ p.p.m.). As previously observed, $\mathbf{1K}$ reacts with water under the release of one equivalent $H_2$ and KOH to give the dark brown neutral complex [Ru(trop$_2$dad)] $\mathbf{2}$ when heated at 60 °C for several hours. Complex $\mathbf{2}$ could not be isolated; however, it is unequivocally characterized by NMR. The electronic structure is best described as a mixture of two resonance forms with either Ru(0) and a neutral diazadiene ligand, [trop-N = CH–CH = N-trop], or as Ru(II) complex of the

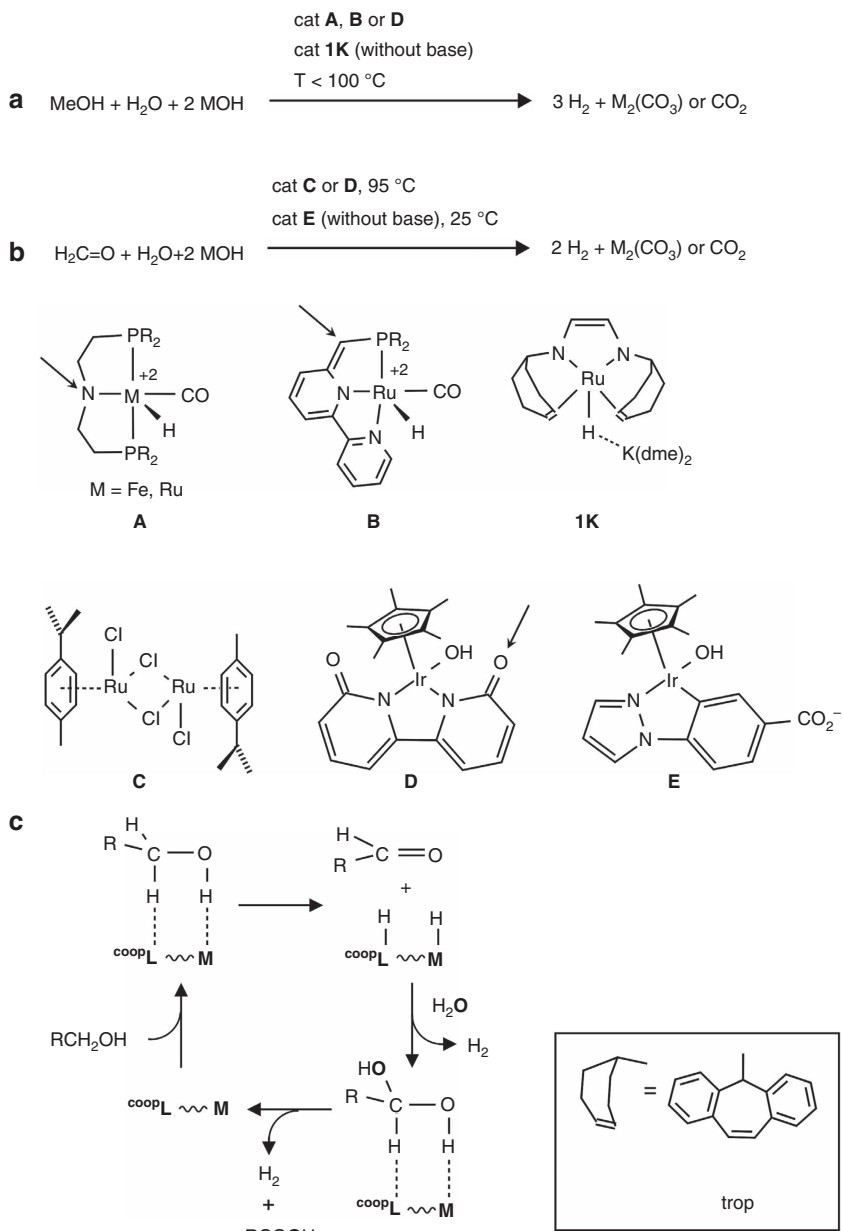

**Figure 1 | Homogeneously catalysed reforming reaction of MeOH and formaldehyde.** Catalysed dehydrogenation of (**a**) methanol/water and (**b**) formaldehyde/water mixtures. The structures of representative catalysts are given by general formulas **A**–**E** and **1K** (**b**); arrows indicate cooperative active sites in the ligand backbone. (**c**) Simplified sketch of the catalytic cycle highlighting the addition of water to the aldehyde and dehydrogenation of the acetal involving metal–ligand cooperation (solvent effects are neglected). A drawing of the trop unit is given at the bottom right.

dianionic form [trop-N$^-$-CH=CH-N$^-$-trop]. This 16-electron complex is a strong Lewis acid and immediately reacts with 2-electron donors (PPh$_3$) to give cleanly the purple 18 electron complex **3a**, which exhibits a square pyramidal structure[15]. In order to test whether the new complexes would be catalytically active as dehydrogenation catalysts of aqueous formaldehyde solutions, we first performed a series of stoichiometric model reactions. These are displayed in Fig. 3.

In an initial experiment, **1Aa** was reacted with water (Fig. 3, equation (a)) and the formation of the neutral complex [Ru(trop$_2$dad)] (**2**) occurred in a notably faster reaction (1 h) at 60 °C giving a higher yield (>70%) when compared to the ion pair **1K** (25%). In tetrahydrofuran (THF), complex **2** reacts with the quaternary ammonium formate [$n$Bu$_4$N][HCO$_2$] to give the hydride complex **1Aa** (Fig. 3, equation (b)). This reaction is not

only an alternative synthesis for complex **1Aa** but also demonstrates that **2** is a potent reagent for the conversion of formate to CO$_2$. Because CO is a possible reaction intermediate in dehydrogenation reactions of formaldehyde, **2** was reacted with carbon monoxide under anhydrous conditions in THF. Complex **2** is converted immediately to give orange crystals of **5** and the carbonyl complex **3b** (Fig. 3, equation (c)). The fact that the dinuclear complex **5** with one CO ligand ($v_{CO} = 1912$ cm$^{-1}$) is formed in high yield is very likely a consequence of its insolubility. The structure of **5** was determined by X-ray diffraction methods (*vide infra*) while soluble **3b** was characterized by NMR spectroscopy. When **1Aa** is reacted with an eight-fold excess of formalin (formalin is a 37% aqueous solution of H$_2$CO, which mainly contains the hydrate of H$_2$CO in the form of short chain oligomers, HO(CH$_2$O)$_n$H and methanol as stabilizer, *vide*

**Figure 2 | Synthesis of penta-coordinated hydride Ru complexes 1 and neutral complex 3.** A drawing of the trop unit and dibenzo-18-crown-6 (db18-C-6) is given at the bottom.

*infra*), gas evolution is observed immediately and the colour of the reaction mixture changes from dark brown to orange. Analysis of the reaction mixture after a few minutes indicated the formation of formic acid (18%, ratio relative to formaldehyde), the yellow *zero*-valent Ru(0) complex **4** (ref. 15), which contains a hydrogenated tropNH-CH$_2$-CH$_2$-NHtrop ligand (41%) and the orange dimeric complex **5H$_2$** (10%), which has a similar structure as **5** with one CO ligand ($v_{CO} = 1,918$ cm$^{-1}$), the difference being the dangling trop unit containing a hydrogenated (CH$_2$CH$_2$)$_{trop}$ unit (Fig. 3, equation (d)). Furthermore, at least three different hydride complexes showing signals in the range $\delta = -5.7$ to $-7$ p.p.m. in the $^1$H NMR spectrum are observed as further minor components in the reaction mixture. Finally, the 18-electron complex **3a** was reacted with formalin at room temperature leading to the blue carbonyl complex **6** ($\lambda_{max} = 377$, 566 nm; $v_{CO} = 1,936$ cm$^{-1}$), which like **5H$_2$** contains a non-coordinated saturated (CH$_2$CH$_2$)$_{trop}$ unit (Fig. 3, equation (d)). When **6** is reacted with H$_2$CO and KOH in a water/THF mixture at 60 °C for 12 h, a sluggish reaction occurs. In the aqueous phase, K$_2$CO$_3$ and formate K(HCO$_2$) was detected. Spectroscopic analysis of the organic phase of the reaction mixture indicated the presence of some unreacted carbonyl complex **6** (10%) and the formation of complex **3a** as main product. Additionally, Ph$_3$P=O is detected (ca. 20%), the new complex [Ru(PPh$_3$)(trop$_2$dae)] (**7**) (dae = diaminoethane) (12% isolated yield), small amounts of the complex **1K**, and two other unidentified species. The Ru(0) complex **7** contains coordinated C=C$_{trop}$ units and a hydrogenated NH–CH$_2$–CH$_2$–NH ligand backbone and can be prepared in good yield by reacting [Ru(trop$_2$dad)(PPh$_3$)] with 2–4 atm of H$_2$ at 65 °C for about 12 h. Mixing a solution of the dinuclear complexes **5** and **5H$_2$** in THF with a solution of 10 equivalents KOH in a water/THF mixture at 60 °C resulted in conversion of the CO ligand to potassium formate and carbonate and concomitant formation of a complex mixture of inseparable ruthenium hydride complexes.

The reactions in equations (a)–(f) in Fig. 3 indicate that various ruthenium complexes may be involved in the dehydrogenation of aqueous formulations of formaldehyde. The reaction between the hydride **1Aa** and formalin gives formic acid as dehydrogenation

product (equation (d)) while **2** is involved in the decomposition of formate (equation (b)). The isolation of the hydrogenated Ru(0) complex **4** indicates that both the metal and the ligand may be involved in the dehydrogenation, while the conversion of formate to CO$_2$ and **1Aa** could be a metal-centred process. Carbonyl complexes may also be formed. Note that in these cases hydrogenolysis of the bonds between C=C$_{trop}$ and the Ru centre could occur to give non-coordinated aliphatic (CH$_2$CH$_2$)$_{trop}$ groups. Under harsher reaction conditions, the CO groups are converted with KOH to formate and/or carbonate (water gas shift reaction) while the (CH$_2$CH$_2$)$_{trop}$ groups are at least partially dehydrogenated back to coordinating C=C$_{trop}$ groups. The reversible hydrogenation/dehydrogenation of these olefinic groups has been previously observed by us with Ir-trop-type complexes[38,39].

The molecular structures of **1Aa**, **1Ab**, **5**, **5H2**, **6** and **7** were determined with X-ray diffraction methods using single crystals. Structure plots of **1Ab**, **5**, **6** and **7** are given in Fig. 4a–d, the other structures are shown in Supplementary Figs 14 and 17. Selected bond parameters are given in Table 1 with additional previously reported data for comparison[15]. The structures of [$n$R$_4$N][RuH(trop$_2$dad)] **1Aa** and **1Ab** show no close contact between the anion and the cation as seen in **1K** and we assume a similar behaviour in solution (Fig. 4a). Otherwise the structure of the [RuH(trop$_2$dad)]$^-$ anion in the ammonium salts is not significantly different from **1K** and is between a trigonal bipyramid and square pyramid with the trop$_2$dad ligand in its enediamide form coordinated to Ru(II). Both solid-state structures of the dinuclear complexes **5** and **5H$_2$** are relatively similar (Fig. 4b). In both dimers, the ruthenium centres reside in a slightly distorted trigonal bipyramidal coordination sphere. Each of the ligands binds via the lone pairs at the nitrogen centres to one Ru centre ($\sigma,\sigma$-coordination) and one pair of $\pi$ electrons of the imine function coordinates to the second metal centre as bridging ligand ($\sigma^2$-N', $\mu^2$-N, $\eta^2$-C=N coordination). These $\eta^2$-C-N units show significantly longer distances (1.425(10)–1.446(10) Å) with respect to the only $\sigma$-N coordinated imine (1.268(10)–1.321(10) Å) (the C=N distance in the free ligand is 1.264(2) Å). In contrast to the anionic hydride [RuH(trop$_2$dad)]

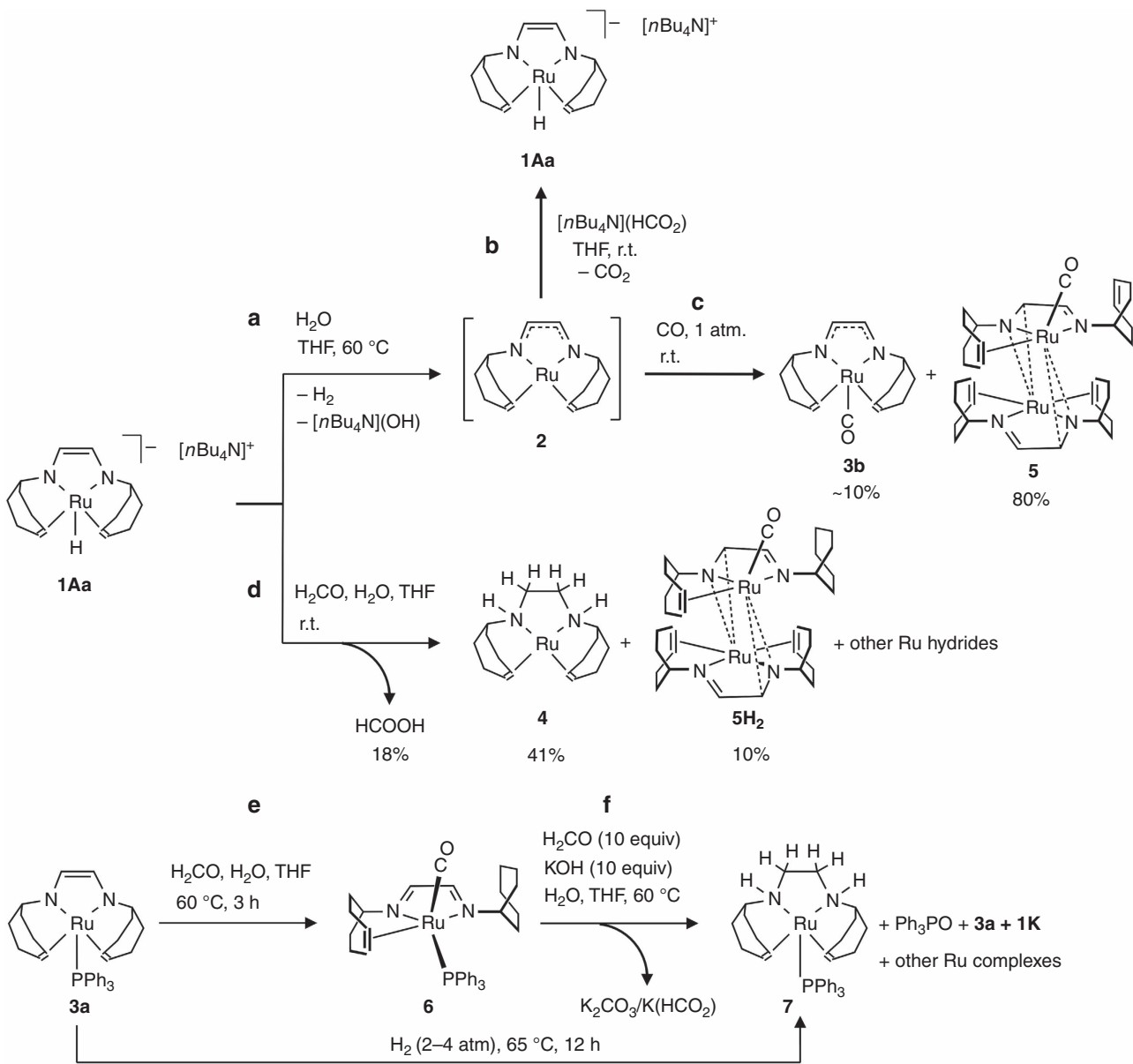

**Figure 3 | Stoichiometric reactivity of complexes 1Aa and 3a.** (a–c) Synthesis and reactivity of complex 2 towards formate and carbon monoxide. (d–f) Model reactions of complexes **1Aa**, **3a** or **6** with formalin.

in **1K**, **1Aa** and **1Ab**, which have a short $C=C$ bond in the $^-N-C=C-N^-$ ligand backbone ($\sim 1.37$ Å avg.), the corresponding C31–C32 and C63–C64 bonds in **5** and **5H$_2$** are significantly longer ($\sim 1.45$ Å avg.), indicating the neutral diazadiene $N=C-C=N$ form of the ligand. The Ru–Ru bond distances are 3.292(1) and 3.293(1) Å and exclude a metal–metal bond. As seen in most trop-type complexes, the coordinating $C=C_{trop}$ bond (C4–C5, C19–C20 and C36–C37) are 0.09 Å (avg.) longer than in the free ligand (1.336(2) Å)[15]. These data indicate that **5** and **5H$_2$** are both best described as Ru(0) complexes, each with strong stabilization of the low-valent metal centres by the ligand through $\pi$-back donation into the $C=N$ and $C=C$ units. All the Ru–N distances in the divalent ruthenium complexes **1** are in the range 1.963(3)–1.978(5) Å. In comparison, an elongation is observed for the Ru1–N1/2 and Ru2–N3/4 bonds (0.13 Å in average) in the zero-valent complexes **5** and **5H$_2$**. In complex **6** (Fig. 4c), the Ru centre resides again in a distorted trigonal bipyramidal coordination sphere and the bond parameters of the N–C–C–N unit indicate that the electronic

structure is best described as a mixture of resonance structures with $^-N-C=C-N^-$ and $N=C-C=N$ units and Ru in the oxidation states of $+ II$ and 0, respectively. That one of the $C=C_{trop}$ units in the ligand became hydrogenated to give a $CH_2-CH_2$ group in **6** (equation (e), Fig. 3) is clearly indicated by the long C19–C20 bond (1.522(5) Å). Figure 4d shows the structure of the reduced complex $[Ru(PPh_3)(trop_2dae)]$ **7** with a distorted trigonal bipyramidal structure. The PPh$_3$, one $C=C_{trop}$ unit, and N2 are located in the equatorial plane while N1 and the remaining $C=C_{trop}$ unit occupy the axial positions. The long C31–C32 bond (1.502(3) Å) shows that the $C=C$ bond in the trop$_2$dad ligand became hydrogenated. The Ru1–N1/2 bond distances (2.189(2) Å avg.) are the longest among the presented ruthenium complexes and are in the same range as those observed in the previously reported ruthenium(0)-dae complex **4**. The Ru centre in **7** has a formal oxidation state of zero. In the $^{13}C$ NMR spectra, large coordination shifts $\Delta(\delta)^{13}C > 70$ p.p.m. indicate strongly bound olefins to low-valent Ru centres (see Supplementary Table 1). The CO ligands in the carbonyl complexes **5**, **5H2** and **6**

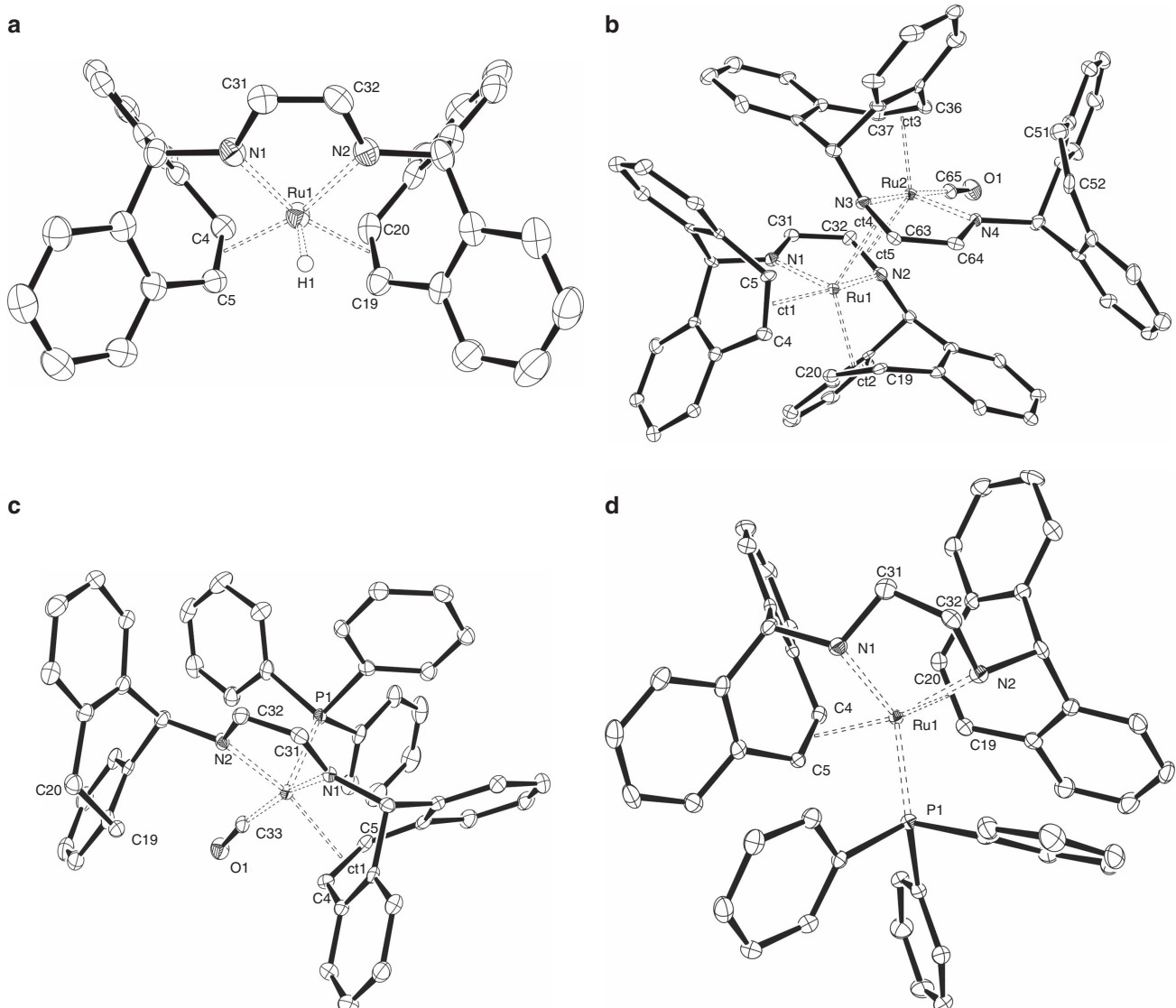

**Figure 4 | ORTEP plots of the complexes.** ORTEP plots of complexes (**a**) **1Ab**, (**b**) **5**, (**c**) **6** and (**d**) **7** as determined by X-ray diffraction studies. Ellipsoids are shown at 50% probability. One molecule of co-crystallized [NBu$_4$]Br and hydrogen atoms, with the exception of H1 in **1Ab** are omitted for clarity.

show resonances in the $^{13}$C{$^1$H} NMR spectra in a narrow window between $\delta = 201.8$ and $\delta = 208.5$ p.p.m. The CO stretching frequencies seen in the IR spectra decrease by $> 200\,cm^{-1}$ compared to free CO. These spectroscopic and structural data show that the ligand trop$_2$dad is electronically remarkably flexible and accommodates Ru between the oxidation states 0 and $+$II. Furthermore, the ligand undergoes reversible hydrogen uptake and release that occurs at both the C=C$_{trop}$ and the N–C=C–N units.

**Catalytic experiments.** Formaldehyde in water forms methanediol (HOCH$_2$OH), which can polymerize to poly(oxomethyleneglycols). The kinetics are strongly dependent on the temperature and pH of the aqueous solution[40,41]. The dehydrogenation of aqueous formaldehyde solutions can proceed following different pathways: (i) The Cannizzaro reaction[42], simplified as $2H_2C=O + H_2O \rightarrow HCOOH + H_3COH$, followed by catalytic dehydrogenation of methanol and formic acid as previously reported[9,15]. (ii) The decarbonylation of formaldehyde according to $H_2CO \rightarrow H_2 + CO$ followed by a water–gas shift

reaction, $CO + H_2O \rightarrow CO_2 + H_2$. (iii) Direct dehydrogenation of methanediol according to $H_2C(OH)_2 \rightarrow HCOOH + H_2$ followed by decomposition of formic acid, $HCOOH \rightarrow CO_2 + H_2$. Our previous studies[15] showed that base is required to convert the hydrogenated diamine trop$_2$dae ligand as seen in **4** and **7** back to the unsaturated diazadiene trop$_2$dad ligand as observed in **1Aa** or **3a**. The presence of base will also drive the reactions (i)–(iii) by forming potassium carbonate as product. THF solutions of complexes **1K**, **1KC**, **1Aa**, **1Ab**, **3a** and **4–7** were tested in the catalytic decomposition of various formaldehyde/water mixtures at 60 °C using a reflux condenser under an inert atmosphere of argon. The progress of the catalytic conversion in this biphasic system was followed by real-time volumetric measurements of released H$_2$ gas. If not noted otherwise, each reaction was repeated three times and averaged data are given in Table 2 (experimental error of $\pm 5\%$). The catalytic reforming of aqueous formalin ($c_0 = 0.47\,M$) was investigated under various conditions (entries 1–13). The highest TON and TOF are achieved with the anionic Ru(II) hydride complexes of type **1** and the neutral Ru(0) complex **4**. Catalyst loadings can be as low as 0.4 mol%. As expected, the activity correlates with the concentration of base

**Table 1 | Comparison of selected bond distances (Å) of complexes 1-7 with previously reported data.**

| Complex | C＝C$_{backbone}$ | C＝N$_{backbone}$ | Ru-N1 Ru-N2 | Ru-ct | Ru-ct$_{backbone}$ | C＝C$_{trop}$ |
|---|---|---|---|---|---|---|
| **1K**[*] | 1.377(8) | 1.348(6), 1.355(8) | 1.963(3) 1.978(5) | 2.034(5) 2.033(5) | — | 1.439(7), 1.433(6) |
| **1Aa**[†] | 1.371(4) | 1.352(5), 1.361(5) | 1.965(3) 1.959(3) | 2.027(4) 2.025(3) | — | 1.435(6), 1.443(6) |
| **1Ab** | 1.371(6) | 1.360(5), 1.364(5) | 1.977(3) 1.976(3) | 2.031(4) 2.047(4) | — | 1.448(4), 1.428(6) |
| **3a**[*] | 1.381(7) | 1.344(6), 1.351(6) | 2.007(3) 1.976(4) | 2.049(4) 2.099(5) | — | 1.444(7), 1.410(7) |
| **4**[*] | 1.522(3) | 1.487(3) | 2.121(2) | 1.970(2) | — | 1.453(3) |
| **5**[‡] | dad-A 1.459(10) dad-B 1.436(9) | dad-A 1.293(12), 1.446(10) dad-B 1.425(10), 1.280(11) | (A) 2.051(6) 2.107(7) (B) 2.095(7) 2.144(6) | (A) 2.108(6) 2.056(6) (B) 2.040(6) | Ru1) 2.038(5) Ru2) 1.985(5) | (A) 1.418(12), 1.439(10) (B) 1.430(11), 1.358(13) |
| **5H$_2$**[‡] | dad-A 1.441(10) dad-B 1.449(10) | dad-A 1.430(9), 1.321(10) dad-B 1.429(9), 1.268(10) | (A) 2.099(7) 2.058(6) (B) 2.110(7) 2.175(6) | (A) 2.067(3) 2.109(4) (B) 2.046(3) | Ru1) 2.022(3) Ru2) 1.988(4) | (A) 1.438(9), 1.399(5) (B) 1.452(9), 1.528(8) |
| **6** | 1.388(5) | 1.320(5), 1.339(5) | 2.038(3) 2.059(3) | 2.068(2) | — | 1.445(5), 1.522(5) |
| **7**[§] | 1.502(3) | 1.494(3), 1.474(3) | 2.175(2) 2.202(2) | 2.052(2) 2.060(2) | — | 1.422(3), 1.459(3) |

[*]See ref. 15.
[†]Complex **1Aa** co-crystallises with one [NBu$_4$]Br molecule.
[‡]dad-A and dad-B refer to each diazadiene ligand in the dimeric complexes. B is the fragment containing the Ru centre coordinated to the CO ligand. One molecule of thf co-crystallises with complexes **5** and **5H$_2$**.
[§]One molecule of dme co-crystallizes with complex **7**.

(entries 1–3, Table 2). Remarkably, catalytic TON is observed in the absence of base but the reaction solution becomes acidic and the conversion drops to 23%. Catalyst **1K** cannot be recycled. The conversion decreases strongly from 86% H$_2$ in the first run to 12% in the second run and the concomitant formation of an insoluble red precipitate is observed (entry 4). The same observation was made with the dibenzo-18-crown-6 complex **1KC** as the catalyst. But the complexes **1Aa** and **1Ab** with quaternary ammonium cations could be reused and these catalysts allowed for repeated substrate addition to the reaction mixture up to six times with little loss of catalytic activity (entries 5–7). No precipitation of the catalyst was observed and TON values of >1700 could be achieved. After the sixth loading, the activity started to level off and yields dropped to ca. 20%. Full conversion, that is the release of two equivalents of hydrogen, was never achieved. NMR spectroscopic analysis of the solution indicates that there are always small amounts of K(HCO$_2$) present (5–7%), although in stoichiometric reactions formate is converted to CO$_2$ (equation (2), Fig. 3). The gas phase was analysed by gas chromatography using a thermal conductivity detector, which showed no detectable traces of CO (for detailed instrumentation setup and gas chromatographic analysis, see Supplementary Fig. 13). With the fully hydrogenated Ru(0) complex **4** as catalyst (entry 8), equally high activities as with complexes **1** were achieved, which is consistent with our previous suggestion that the ligand cooperates with the metal and participates in C–H activation steps whereby the substrate is dehydrogenated and H$_2$ is transferred to the ligand backbone.

As shown in equation (d) in Fig. 3, complex **1Aa** also decarbonylates formaldehyde and transfers H$_2$ to one of the C＝C$_{trop}$ units and binds CO to the Ru centre forming complex **5H$_2$** under mild acidic conditions (pH = 6). Under optimal catalytic conditions (pH >12), complex **5H$_2$** is able to catalyse the formaldehyde reforming reaction but with significantly lower TON and TOF numbers (entry 9). Similar results were obtained with the dimeric carbonyl complex **5** which contains the decoordinated but unsaturated C＝C$_{trop}$ unit (entry 10). We tested the penta-coordinated phosphane complexes **3a**, **6** and **7** as catalysts for formalin reforming (entries 11–13). The best results were obtained with the complexes [Ru(PPh$_3$)(trop$_2$dad)] **3a** and the *zero*-valent 18 valence configured Ru complex **7**; the latter can be converted to **3a** in the presence of an excess of base. In this reaction the formation of **1K** is also observed while the PPh$_3$ ligand is oxygenated to Ph$_3$PO which is detected by [31]P NMR spectroscopy. As in equation (e) in Fig. 3, Ru(II) complex **3a** decarbonylates formaldehyde in the absence of base and gives complex **6** as the only product. This complex reacts sluggishly under catalytic conditions and shows the lowest performance of all Ru complexes we tested in the formalin-reforming process. The most notable results were achieved in the conversion of aqueous solutions of paraformaldehyde ($c_0$ = 0.47 M), which is a mixture of polyoxymethylenes, HO(CH$_2$O)$_n$H with $n$ = 8–100 repeating units. With catalyst **1Aa**, high conversion (up to 90% H$_2$) and fast gas flow (TOF$_{50}$ > 20,000 h$^{-1}$) is observed under basic conditions. The catalyst can be recycled without significant loss of efficiency (entries 14,15). It is very remarkable that the conversion of aqueous paraformaldehyde can be performed

**Table 2 | Catalytic activity in the decomposition of formaldehyde/water mixtures by Ru complexes\*.**

$$HO(CH_2O)_nH \rightleftharpoons CH_2(OH)_2 \rightleftharpoons H\!-\!\overset{O}{\underset{}{\parallel}}\!-\!H + H_2O \xrightarrow[60\,°C]{0.4\,mol\%\,[Ru]\ KOH\,(x\,equiv)} K_2CO_3 + 2\,H_2$$

$$H_2O \quad HO(CH_2O)_{n-1}H$$

| Entry | Catalyst | KOH (equivalents) | TOF$_{50}$ (h$^{-1}$)[†] | Total yield H$_2$ (%)[‡] | TON$_{max}$/duration[§] |
|---|---|---|---|---|---|
| 1 | **1K** | — | — | 23 | 115/12 h |
| 2[‖] | **1K** | 2 | 8,109 | 56 | 280/12 h |
| 3[‖] | **1K** | 6 | 17,500 (first load) | 86 | 430/2 h |
| 4 | **1K** | | — (second load) | 12 | 103/2 h |
| 5[‖] | **1Aa** | 6 | 15,101 (first load) | 90 | 450/12 min |
| 6[¶] | **1Aa** | 6 | 12,000 (sixth load) | 59 | 1,787/20 min |
| 7[‖] | **1Ab** | 6 | 13,520 | 81 | 405/15 min |
| 8[‖] | **4** | 6 | 17,000 | 90 | 450/2 h |
| 9 | **5H$_2$** | 6 | 6,000 | 68 | 340/4 h |
| 10[‖] | **5** | 6 | 7,500 | 75 | 375/4 h |
| 11[‖] | **3a** | 4 | 3,537 | 58 | 290/4 h |
| 12[‖] | **6** | 4 | 750 | 69.5 | 347/4 h |
| 13[‖] | **7** | 4 | 4,091 | 65 | 325/4 h |
| 14[#] | **1Aa** | 6 | 29,764 (first load) | 90 | 450/15 min |
| 15[\*\*] | **1Aa** | 6 | 22,000 (second load) | 85 | 765/15 min |
| 16[††] | **1Aa** | 6 | 805 | 92 | 460/2 h |
| 17[‡‡] | **1Aa** | 6 | — | <5 | —/12 h |

\*Reaction conditions: formaldehyde (1.0 mmol) $c_0 = 0.47$ M, 0.4 mol% [Ru] at 60 °C in water/THF (10:1).
[†]TOF values after 50% conversion (1 equivalents H$_2$ released per formaldehyde unit).
[‡]Yield considering 2 equivalents H$_2$/equivalent HCOH.
[§]TON = mmol H$_2$ released per mmol [Ru].
[‖]Values are an average of three catalytic runs.
[¶]Final value after the sixth addition of HCOH aq. to the reaction mixture of entry 5.
[#]Paraformaldehyde (1.0 mmol) $c_0 = 0.47$ M, 0.4 mol% [Ru] at 60 °C in water/THF (10:1).
[\*\*]Final value after the second addition of HCOH to the reaction mixture of entry 14. Average of three runs.
[††]Same conditions as in entry 14 under CO$_{(g)}$ atmosphere. Average of three runs.
[‡‡]Same conditions as in entry 16 under air. Average of two runs.

under an atmosphere of CO although with lower activity (entry 16). Only when the catalytic reaction is performed in the presence of oxygen (that is, without deoxygenating the solvents and/or under air), no activity is observed (entry 17).

We considered the possibility that formaldehyde disproportionates in a Cannizzaro reaction as shown in (i). In that case, the complexes listed in Table 2 may merely catalyse the dehydrogenation of formic acid and methanol. When formaldehyde (0.5 M) is heated with 6 equivalents of KOH in D$_2$O at 60 °C (without any Ru catalyst), which corresponds with the conditions used in the Ru-catalysed reactions, only 6% conversion is obtained after 15 min. The conversion increases to about 30% after 12 h. It is known that the Cannizzaro reaction proceeds with high efficiency at higher concentration[43] and indeed with a 5 M formaldehyde solution at pH = 14 and 60 °C, >90% conversion to formate and MeOH is achieved. When the catalyst **1Aa** is added to this mixture and heated to 60 °C, only 22% of H$_2$ is evolved. This corresponds approximately to the expected amount of H$_2$ from the decomposition of formic acid (25%). Analysis of the reaction mixture by NMR reveals traces of formate (<2%) and 97% of methanol. Furthermore, a rather rapid decomposition of the Ru complex to an insoluble red solid was observed. Hence, methanol is not converted under these conditions, which is in contrast to our previous report where complex **1K** was found to convert methanol/water mixtures but at much lower base concentrations and higher temperatures[15]. Note also that generally the efficiency of the catalysis decreases with increasing formaldehyde concentration >0.5 M in water (Supplementary Table 2). The higher catalytic efficiency using diluted formaldehyde solutions was also observed previously by Prechtl *et al.*[26]. Hence we conclude that the catalytic reactions proceed via direct dehydrogenation of formaldehyde/methanediol, and the Cannizaro reaction plays only a minor role under the applied reaction conditions.

**Mechanistic DFT study**. The experimental stoichiometric reactions and catalytic studies clearly indicate that there are several ruthenium species formed under catalytically relevant reaction conditions, and likely several mechanisms are operative in the conversion of aqueous formaldehyde solutions to carbonate and hydrogen under basic conditions (*vide supra*). A complete survey of all possible reaction pathways is far beyond the scope of this study. We therefore focussed on the possibility that the NCCN backbone of the ligand participates in key steps of the dehydrogenation reaction and we assumed that the reaction actually proceeds via dehydrogenation of methanediol[44]. The possible decarbonylation of formaldehyde was not investigated here because the experimental data show that this reaction pathway is likely less favourable or perhaps even a catalyst deactivation pathway. Likewise, the possible disproportionation of formaldehyde to formic acid and methanol was not further investigated. This reaction may be a side reaction but is unlikely to be the major reaction pathway. The experimental data provide some insight that guided our mechanistic DFT studies. Complexes **1** and **4** give rise to much higher TOFs and TONs than obtained with any of the other complexes isolated, and as such, species **3b**, **5**, **5H$_2$** and **6** are not likely to be catalytically relevant (Table 2). Quantitative and fast formation of neutral complex **2** and H$_2$ from the anionic hydride precursors **1K** and **1Aa** upon reaction with water under catalytically relevant conditions is of key importance. As is the formation of the catalytically highly active neutral species **4** with a fully hydrogenated dad-backbone upon reaction of **1Aa** with water and formaldehyde at room temperature (see Fig. 3). This, in combination with substrate binding being comparatively disfavoured for the saturated (18 valence electron) anionic hydride species **1K** and **1Aa**, makes it most plausible to propose that the neutral, unsaturated (16 valence electron) species **2** and **4** are involved in the TON. As such, we focussed

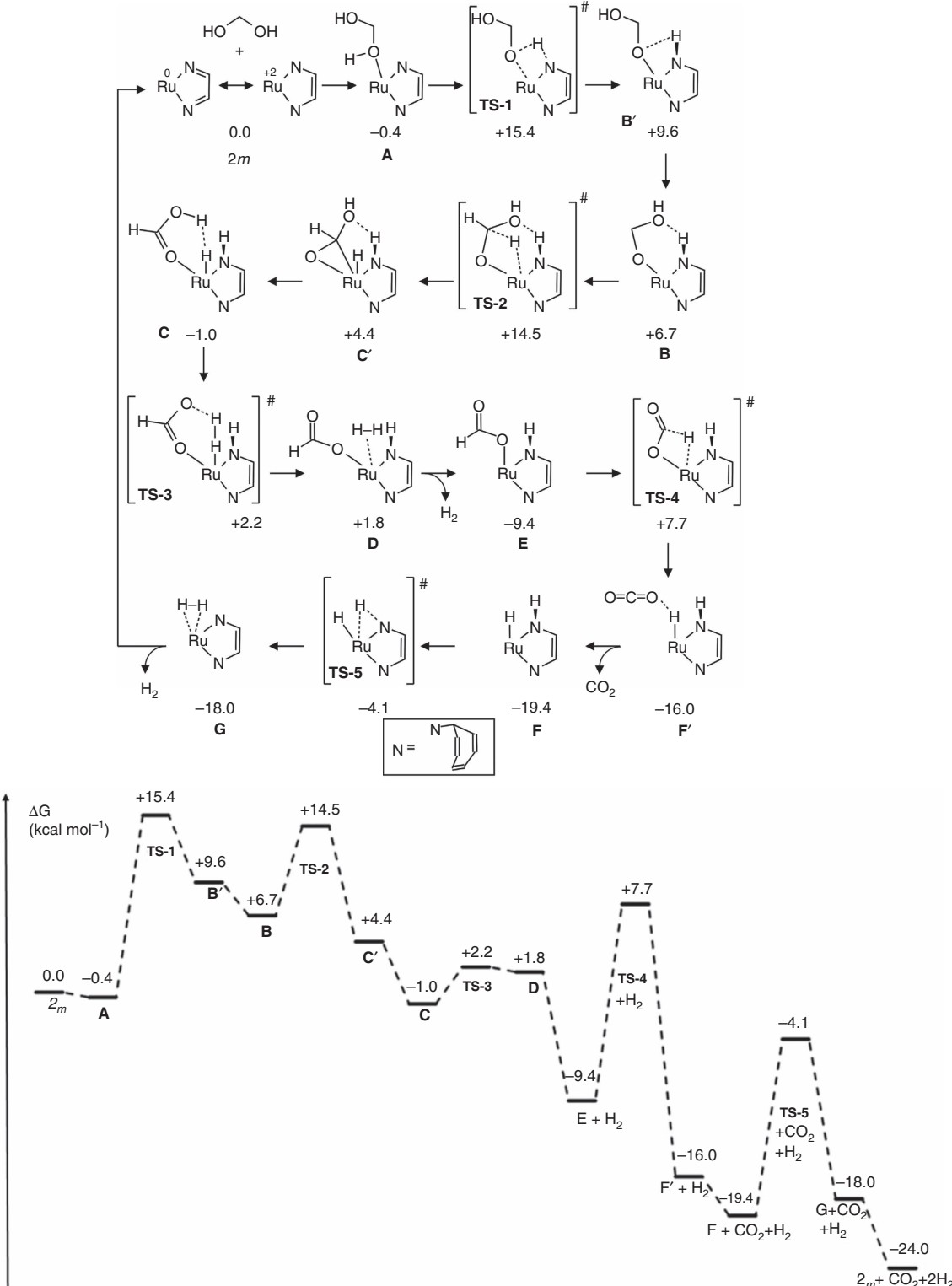

**Figure 5 | Computed pathway for methanediol dehydrogenation catalysed by complex $2_m$.** Calculated pathway (Turbomole, DFT-D3 (disp3), BP86, def2-TZVP) along with their relative free energies ($\Delta G°_{298K}$ in kcal mol$^{-1}$). All energies (also the transition states) are relative to the starting materials (complex $2_m$ + methanediol).

our DFT study on these neutral species. We chose simplified models of complexes **2** and **4** (referred as $2_m$ and $4_m$, respectively) as catalysts that contain no annulated benzo groups in the trop moiety. DFT methods were used for the calculations and the Minimum Energy Reaction Pathway (MERP) with $2_m$ as catalyst is shown in Fig. 5. The MERP with the fully hydrogenated

complex $4_m$ (and its related isomers) is given in Supplementary Figs 23 and 25 and has a very similar energy profile.

The catalytic cycle starts with exergonic formation of methanediol adduct **A**, followed by proton transfer from the alcohol moiety of the substrate to one of the dad-backbone nitrogen atoms via transition state **TS-1**, thus producing

complex **B′**. The overall deprotonation step is slightly uphill with a barrier of $+15.4\,\mathrm{kcal\,mol^{-1}}$ (the barrier of this process may be overestimated, as proton transfer in solution is most likely solvent assisted). Complex **B′** then rearranges to hydrogen-bond-stabilized complex **B**, which by β-hydride elimination subsequently converts to complex **C′** via **TS-2** at a transition state energy of $+14.5\,\mathrm{kcal\,mol^{-1}}$. This low barrier is in marked contrast to a recent computational study, which claims that β-hydride elimination from a related metal-bound alcohol/alcoholate adduct is not feasible[35]. Note that, in the pathway involving the fully hydrogenated complex $4_m$ (Supplementary Fig. 23), the computed barrier for hydride transfer from the substrate to the metal is even lower $(+3\,\mathrm{kcal\,mol^{-1}})$. In this particular case, this process does not proceed via a classical β-hydride elimination but instead involves an ion-pair polarized transition state stabilized by two hydrogen bonds (**4-TS-1**; Supplementary Fig. 23). Subsequent rearrangement of **C′** produces **C**, which is preorganized for protonation of the hydride by the coordinated formic acid moiety to produce $H_2$. Complex **C** is clearly stabilized by a dihydrogen bond (hydride–proton interaction). Formation of dihydrogen adduct **D** via **TS-3** has a rather low barrier $(<+5\,\mathrm{kcal\,mol^{-1}})$ and subsequent release of $H_2$ is essentially barrier-less thus producing formate complex **E**. Direct β-hydrogen elimination from the formate ligand in **E** via **TS-4** produces ruthenium hydride complex **F′**. This process has the highest barrier $(+17\,\mathrm{kcal\,mol^{-1}})$ in the catalytic cycle and seems to be the TOF-limiting step for catalyst $2_m$. The computed overall barrier for the reaction seems to be somewhat low for a reaction requiring heating in the experimental reactions. The apparently underestimated barrier might be due to the simplified ligand used in the computational studies (truncated trop moiety), unaccounted explicit solvation effects in the gas-phase DFT calculations and/or limitations of the functional used. However, addressing all these issues is beyond the scope of the present paper, which aims at providing a qualitative picture of the most likely pathways occurring at the ruthenium centre. The resulting formation of **F′** is quite exergonic, and loss of $CO_2$ from **F′** to form **F** is further downhill on the energy landscape. Proton transfer from the ligand to the metal across the Ru–N bond of **F** via transition state **TS-5** leads to formation of $H_2$ complex **G**, which readily loses $H_2$ to complete the catalytic cycle. The MERP computed with catalyst $4_m$ (Supplementary Fig. 23) involves ion-pair polarized, hydrogen-bond-stabilized intermediates (and transition states) in many of the computed steps but has comparably low barriers for all individual reaction steps. As such, the computed pathways for methanediol dehydrogenation by both catalysts **1Aa**, which rapidly converts to **2** under the experimental conditions, and **4** provide viable pathways for formaldehyde dehydrogenation. It is quite likely that both are used and contribute to the observed catalytic activity. In both pathways, hydride migration from the substrate to the metal are key steps in the catalytic cycle to produce $H_2$, and in both mechanisms metal–ligand cooperativity plays an important role.

## Discussion

The ruthenium complexes of type **1** with the $[\mathrm{RuH(trop_2dad)}]^-$ anion, as well as the neutral Ru(0) complex **4**, catalyse the conversion of alkaline aqueous solutions of formaldehyde (as formalin or paraformaldehyde) into hydrogen and carbonate at 60 °C in a biphasic reaction system with unprecedented high efficiency. We assume that methanediol is the substrate that is converted by the ruthenium complexes as catalysts. The faster rate of formation of $H_2$ from aqueous paraformaldehyde might be due to the higher concentration of methanediol in formalin solutions, which consists of higher oligomeric mixtures of methyleneglycols.

The catalytic system contains no phosphanes as ligands, which significantly improves the energy balance of a catalytic system that is designed to deliver hydrogen as energy carrier (the synthesis of phosphanes necessitates the reduction of phosphate rock, which is a highly energy-intensive process that is likely not counterbalanced by the catalyst during its lifetime). It is very likely that water, and not $O_2$, serves as source of oxygen in the final product, and in reactions of organic aldehydes with $O^{18}$-labelled water, $\mathrm{RCH}{=}\mathrm{O^{16}} + \mathrm{H_2O^{18}} \rightarrow \mathrm{RCO^{18/16}O^{18}H} + \mathrm{H_2}$, this hypothesis was proven[15,28]. In aqueous formaldehyde, water is already incorporated into the acetal molecule prone to be dehydrogenated to formic acid. The ruthenium centre switches its oxidation states between zero and $+$II and the ligand very likely participates in various ways in individual steps of the catalytic reaction. The ligand serves not only as a redox non-innocent entity but also participates chemically in the dehydrogenation of methanediol and its oligomers. DFT calculations indicate that both the metal and the ligand play important roles and the activation of the substrate occurs via addition across the polar Ru–N bond followed by intramolecular β-hydrogen elimination steps at the metal centre. These reaction steps are meanwhile well-established transformations in organometallic chemistry. The stoichiometric experiments show that hydrogen may not only be transferred to the metal centre but can also be stored in various sites of the ligand to give complexes with hydrogenated ligands such as in the Ru(0) amine complex $[\mathrm{Ru(trop_2dae)}]$ **4**, which stores two equivalents of $H_2$. Likely via 1,2-hydrogen shifts from the ligand to the metal centre, the hydrogen content of the ligand is then decreased and $H_2$ is released. The olefinic-binding sites may be hydrogenated as well and displaced from the metal centre. But because the $CH_2$-$CH_2$ groups remain in proximity of the metal centre, these reactions are also reversible. Finally, CO complexes have been detected which indicate that the ruthenium amine/imine complexes may likewise decarbonylate (hydrated) formaldehyde and catalyse the conversion of CO to carbonate in course of a water–gas shift reaction. This allows to run the formaldehyde–water–gas shift reaction even under an atmosphere of CO, although with diminished efficiency. The catalysts reported here may be considered as 'catalytic chameleons' in the sense that they adapt to their environment and thereby show a remarkable ability to adjust to various reaction conditions.

## Methods

**Synthesis of $[\mathrm{NBu_4}][\mathrm{Ru(trop_2dad)}]$ (1Aa).** $H_2O$ is added (20 equivalents) to a solution of complex **1K** (1.0 equivalents, 11 mM) in THF and the mixture is heated for 6 h at 60 °C under an argon stream, while the initial brown solution turns gradually to dark red. After filtration, all volatiles are removed. The obtained residue is washed with $n$-hexane and dried under vacuum to give pure complex **2** as dark red solid. $[\mathrm{Bu_4N}][\mathrm{HCO_2}]$ (1.1 equivalents) is added to a solution of complex **2** (1.0 equivalents, 9.3 mM) in THF and the mixture is stirred at room temperature for 1 h, causing a colour change from red to violet. The solution is filtered through a syringe filter (50 μm porosity). The obtained clear filtrate is layered with $n$-hexane and cooled to $-32$ °C. After 2 days, air-sensitive burgundy red crystals of complex **1Aa** are isolated by filtration and dried in a stream of argon (51% yield).

**Dehydrogenation of aqueous formaldehyde and paraformaldehyde.** A 25 ml two-neck round-bottom flask is connected to a reflux condenser with argon inlet/outlet, which is coupled to a water-filled gas burette, while the second neck of the flask is capped with a septum. The reaction vessel is purged with argon–vacuum cycles for 20 min in order to remove air and moisture. A degassed solution of formaldehyde (75 μl of a 37 wt% aqueous solution, 1.0 mmol, 1.0 equivalents) or suspension of paraformaldehyde (30.0 mg, 1.0 mmol, 1.0 equivalents) in $H_2O$ (2 ml) is added followed by the required amount of base and the mixture is heated at 60 °C. After equilibration, the required amount of ruthenium complex in THF (200 μl) is added with a syringe. The mixture is stirred vigorously and the volume of liberated gas is recorded periodically until gas evolution ceases. Released hydrogen was quantified by recording its volume displacement in the eudiometer and correcting its volume for water content.

**Data availability.** Atomic coordinates and structure factors for the reported crystal structures have been deposited in the Cambridge Crystallographic Data

Centre under deposition number CCDC-1502942 (**1Aa**), 1502945 (**1Ab**), 1502948 (**5**), 1502951 (**5H2**), 1502952 (**6**) and 1502953 (**7**). These data can be obtained free of charge from the Cambridge Crystallographic Data Centre via www.ccdc.cam.ac. uk/data_request/cif. Detailed experimental procedures, characterization of compounds and the computational details can be found in Supplementary Figs 1–27, Supplementary Tables 1–9 and Supplementary Methods and are also available from the corresponding author upon reasonable request.

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

## Acknowledgements

This work was supported by the Schweizer Nationalfonds (SNF), Eidgenössische Hochschule Zürich, the FOM-NWO-Shell Computational sciences for energy research initiative (project 13CSER003) and the RPA Sustainable Chemistry of the University of Amsterdam.

## Author contributions

H.G., B.d.B and M.T. directed and conceived this project. M.T., R.E.R.-L. and B.P. conducted the experimental work. V.S. conducted the computational work, guided by B.d.B. All authors discussed the results and wrote the manuscript.

## Additional information

**Competing interests:** The authors declare no competing financial interests.

