## [Peer review file · Nature Communications]

Reviewers' comments:

Reviewer #1 (Remarks to the Author):

This manuscript reports a combined experimental and theoretical study of trop2dad Ru catalyzed conversion of formaldehyde and water to H₂ with the presence of base. This catalytic system is the fastest acceptorless formaline dehydrogenation reported so far. Considering the importance of the dehydrogenation of formaldehyde and the high efficiency of this new catalytic reaction, this manuscript will be interested in a wide readership. I am willing to support its publication in Nature Communications. However, I would like to see the following concerns to be addressed before it can be further considered.

1. In the mechanistic study, authors declare that the reactions catalyzed by model structures 2m and 4m have very similar energy profiles. I was wondering what is the relative free energy between 2m and 4m? If 4m is more stable, what is the barrier for the conversion of 2m to 4m in the reaction with the existence of H₂ and the assistance of water molecule for H₂ cleavage?
2. In Figure 5, intermediate F is the resting state with the lowest relative free energy of 19.0 kcal/mol, while transition state TS-1 has the highest relative free energy of 15.4 kcal/mol. Therefore, the total free energy barrier of this catalytic reaction is 34.8 kcal/mol, which seems too high for a reaction at 60 °C.
3. In Figure 5, the reaction starts with methanediol as the reactant. What is the relative free energy for the formation of methanediol from formaldehyde and water? This might be an important step in the catalytic conversion of whole formaldehyde and water to H₂.
4. In addition to list the relative energies in Figure 5, I suggest authors to provide a free energy profile with reaction coordinate and relative energies for a clearer description of the reaction mechanism.
5. Please specify clearly which kind of energy, electronic energy, enthalpy, or free energy, is the energies along with the atomic coordinates.
6. The names of authors of Ref. 36 seem redundant.

Reviewer #2 (Remarks to the Author):

Dear authors,

I read with interest your manuscript entitled "Water as Oxygen Transfer Reagent in a Homogeneously Catalysed Conversion of Formaldehyde to H₂", the provided supporting information, and I checked the cited key-references about the hydrogen generation from formaldehyde in water.

In general, H₂ generation at low temperature using small hydrogen-rich molecules is a topic of general importance for the scientific community and also mechanistic studies are very important to achieve optimised catalysts for H₂ generation. The results are well presented and prepared, and the analytical details are of high standard. However, some aspects must be clarified previous to publication and undoubtedly revisions are required.

Please consider the following comments for the revision of your manuscript:

- The introduction starts with water as oxygen source for oxygen-transfer reactions. This has been discussed and analyzed by isotope-labelling experiments with labelled water during the alcohol dehydrogenation via aldehydes in detail in *Nat. Chem.*, 5, 122–125,(2013) doi:10.1038/nchem.1536). In 2014, another group observed and described this oxygen transfer reaction in aqueous formaldehyde solution again using oxygen-labelled molecules (*Nat. Commun.* 2014, 5, Article number: 3621, doi:10.1038/ncomms4621). So this, aspect is not really new. Indeed, while reading the manuscript, I found you related the incorporation of oxygen from water into the well-known carbonate formation. I do not know the original article, but a patent filed in the late 18th century (by Schweppes) described the saturation of water with CO₂, forming aqueous carbonic acid (H₂CO₃), respectively carbonate under basic conditions. In this regards, I feel you over-sell the non-catalytic phenomena of oxygen-transfer from water to CO₂ (yielding sparkling water), the CO₂ is formed during the formaldehyde decomposition, and than redissolved in the water giving carbonate at the applied basic conditions (1-3M KOH for example!). I do not really understood, why have you used the oxygen-transfer from water to carbonate in alkaline media as 'cliff-hanger' in the title of this fine work.

- In your manuscript (introduction and results) and the description of the state of the art about methanol and formaldehyde dehydrogenation, I recognised you mix two different information in an inappropriate way:

Adding a base to a system does not imply directly that the reaction has been conducted under basic conditions and the base concentration plays an important role in the system in regards to the pH value. For example the system described for methanol dehydrogenation (*Nature* 495, 85–89 (2013); Figure 1a, catalyst A), is conducted in a MeOH/H₂O mixture with 8M KOH, unquestionable this is strongly basic (pH = 14); even a solution of 1M KOH/water gives a pH of 14. In contrast in Figure 1b, applying catalyst C, the cited reference, describes the addition of base to "buffer" the in situ formed formic acid and HCl, the reported pH is below 7. Thus, these are no basic conditions at all.

- You probably overlooked one article about formaldehyde dehydrogenation, because you wrote on page 3, that the best catalyst gives a TOF of 170 per hour and a TON of 188. A fast check on Web of Science showed that there is article of 2016 reporting a TOF of 3142 per

hour and a TON of 700 under near neutral conditions (Green Chem. 2016, 18, 1469-1474) which is almost 20-times higher than the cited one.

Now I must point out the major problems of this study which have been not considered at all:

On page 12 the author state correctly the "... kinetics are strongly dependent on the temperature and pH of the solution." Now I have mention again that the base concentration is very important in this particular case. From organic chemistry textbooks we learn that aldehydes undergo disproportionation to the corresponding alcohols and carboxylic acids under basic conditions ($\text{pH} > 11$). This reaction is known as the Cannizzaro reaction (https://en.wikipedia.org/wiki/Cannizzaro_reaction; S. Cannizzaro, 1853: doi:10.1002/jlac.18530880114). The authors applied an excess of 2, 4, 6 eq. of KOH to aq. formaldehyde in most reactions (refer ESI and discussion). The given concentrations are equal to 1-3M solution of KOH in aq. formaldehyde. Undiscussable these are strongly alkaline conditions with a pH 14. Heating aqueous formaldehyde to to 60°C at pH 14 results in the formation of methanol and formic acid. This could be easily checked with a blank experiment and analysis by $^1\text{H-NMR}$ to confirm the well known Cannizzaro reaction (it would be very surprising if the authors could reveal that the Cannizzaro reaction do not take place in basic media!). Luckily, the authors applied a catalyst which is known to be active also for methanol dehydrogenation (Nat. Chem. 2013, 5, 342–347.) - more about this aspect on a later point. Considering the Cannizzaro reaction and the obvious formation of methanol and formic acid, the catalyst dehydrogenates in the first instance the formic acid to CO_2 and H_2 with a TON of >1700 (TOF of 12000 per hour) which is not so fast and there are many publications on formic acid decomposition giving much higher TONs in the range of >100.000 . Interestingly the authors report on page 12 that "Full conversion, that is the release of two equivalents of hydrogen, was never achieved". One equivalent yields from the formic acid decomposition, and some amounts are generated from the methanol dehydrogenation which is probably slower, more over the authors reported residual potassium formate which is not fully decomposed. The H_2 derived from methanol can be estimated by NMR analysis. Most interestingly, the catalyst 1Aa (page 14) decomposed paraformaldehyde with a TOF of >20.000 per hour, yielding 90% of H_2 . This is very good.

- However the proposed reaction pathway based on DFT calculations (Figure 5) is probably more complex. 1. Cannizzaro reaction occurs, 2. formic acid is dehydrogenated to H_2 and CO_2 (respectivel carbonate under basic conditions), 3. methanol is dehydrogenated to formaldehyde and H_2 , 4. the generated formaldehyde from the methanol dehydrogenation undergoes in the next cycle a Cannizzaro reaction again, and so on.... So, the Cannizzaro reaction, in this particular case, is not a dramatic story, because a catalyst has been used which is luckily active for the methanol dehydrogenation. Therefore, the reaction pathways are simply different and a new proposed mechanism is required to be included into the publication - an extended cycle.

- A positive aspect is that the reaction can be run under CO atmosphere (page 15) which opens new possibilities. On the same page they mention the major drawback of their system, besides the strong basic conditions (!; 1-3 M KOH in aq. formaldehyde!), the catalyst is just stable in deoxygenated solvents and in absence of air! Under air and non-deoxygenated water, the catalyst is completely inactive and cannot compete with air-stable catalysts.

- On page 18, the authors state again that carbonate is formed under basic conditions. I would say that this is again a drawback in regards to the accumulation of a salt in the liquid phase, contrary one could argue the purity of the delivered H₂ is higher. However the carbonate must be recycled or separated in some way.

- On page 18, the author state again: "It is very likely that water serves as oxygen transfer reagent and in reactions of organic aldehydes with O¹⁸ 399 labelled water, RCH=O¹⁶ + H₂O¹⁸ → RCO¹⁸/16O¹⁸H + H₂, this hypothesis was proven.⁴⁵" Ref. 45 states: "⁴⁵ In the homogenously catalysed reaction of benzaldehyde with H₂O¹⁸ with complexes 1 and 3a PhCO¹⁸O¹⁸K/ PhCO¹⁶O¹⁸K/ were obtained. These results will be published separately. See also reference 15."

I would say that this has already been proven by means of isotope -labelling experiments that water serves as oxygen transfer reagent, vide supra and refer: Nat. Chem., 5, 122–125,(2013) and Nat. Commun. 2014, 5, Article number: 3621, doi:10.1038/ncomms4621. Please do not use the oxygen transfer from water to aldehyde and water to CO₂ as 'cliff-hanger'. This is not really necessary and already known.

In summary, the paper give some interesting insights about new catalysts for the H₂ generation from formaldehyde under basic conditions in oxygen-free solvent and atmosphere. The report needs a revision to become suitable to be published in this journal - see comments above. The major concerns are the basic conditions which lead to the well-known Cannizzaro reaction pathway. Consequently, the whole reaction pathway is different, but luckily owing to the use of the right catalyst which is capable for methanol dehydrogenation high yields of H₂ are still possible. The authors reached their target, but via an overlooked and unconsidered but well-known reaction pathway.

Reviewer #3 (Remarks to the Author):

While this manuscript is generally well-written and of interest to members of the community, I can not recommend publication in its current form. There are a number of issues with the crystallographic data in the body of the paper as well as in Supplementary Material. In order to perform a thorough review, I attempted to run the 6 CIF files through the standard checkCIF review. Most of the CIFs contained syntax errors which prevented a checkCIF analysis until I made the necessary edits. This should have been done by the authors before submission. In addition, one of the CIFs did not include hkl data which prevented a complete analysis. The data for 1Aa were included in the SI but little detail was provided in the text as to its formulation. The CIF contains 19! partial occupancy Br sites as well as a partial occupancy hexane site including hydrogens. Clearly, this was just an attempt to "mop up" residual electron density in the Fourier map but anything could have been modeled at these sites. Therefore, the formulation for 1Aa has not been unambiguously established from x-ray data...nor was there any attempt to provide it in the CIF. At the very least, a comment should have been included in the CIF with respect to the choice of model. Several of the 6 CIF files generated A and B level alerts in checkCIF. Multiple other alerts were generated by checkCIF but these can generally be read and ignored. However, comments regarding the A and B alerts should be included in the CIFs. For compound 1Ab, the Ru-H distance is unrealistically short (1.19Angstrom)(based on a survey of the Cambridge Database). While the location and refinement of hydrides in the presence of heavy atoms is a well-known crystallographic problem, at the very least, a comment recognizing this anomaly is required.

The Figure captions for Figure 4 contain several errors as well. It's apparent that the final names in the CIFs do not match those in the Figure captions and at some point, the structures were renamed. As such, it makes it quite difficult to follow the authors discussion. For example, the values given in caption 4d for C4-C5 and C19-C20 do not match the values in the CIF file. For Figure 4b, Ru-N and other Ru distances are reported in the caption but no explanation for Ru2-ct5? and no discussion in the text? In addition, Ru - N and Ru-C distances aren't provided for any of the other structures so why include them here? In the discussion, standard uncertainties are not provided on several of the reported values making it difficult to assess the magnitude and significance of the differences.

Clearly, the authors have done a great deal of work in the preparation of this manuscript. I sympathize that the preparation of papers reporting multiple structures is tedious. However, the work here is incomplete. The crystallographic data could do a great deal to support the authors arguments but it needs to be "cleaned up" and better presented. At the very minimum, accurate formulas should be given in the text for each of the structures (including solvent/disorder); i.e. what is 1Aa vs. 1Ab, etc. For the discussion of the metrical structural parameters, it would be much better for the values to be compared in tabular form as opposed to figure captions; i.e. Table S2 would be a valuable addition to the text as opposed to buried in supplementary material. Or, at least, the existence of the table should be referenced in the text. There's a great deal of supportive evidence here but the authors have hidden it and could make it much easier to find.

After significant revision and additional refinement for some of the structures with corrected CIFs, I believe that this paper should be reconsidered for publication.

We would like to thank the referees of this manuscript for the helpful comments and for the rapid review of our manuscript. Our responses to the suggestions for improvement our scientific communication are indicated below. All changes have been highlighted in yellow in the main text and experimental section. New cif files and check cif reports of all new complexes are provided.

Reviewer 1, comment 1:

In the mechanistic study, authors declare that the reactions catalyzed by model structures 2_m and 4_m have very similar energy profiles. I was wondering what is the relative free energy between 2_m and 4_m? If 4_m is more stable, what is the barrier for the conversion of 2_m to 4_m in the reaction with the existence of H₂ and the assistance of water molecule for H₂ cleavage?

Response:

Experimental results indicate that both **2** and **4** can co-exist under catalytic conditions (open vessel, constant removal of hydrogen formed). We expect species **2** and **4** to exist in an equilibrium in the presence of hydrogen. Experimental results also indicate that **4** can be converted to **2** releasing two equivalents of H₂ by heating in presence of base (see main text page 12). In addition to these experimental results, Li and Hall and Yang and co-workers extensively examined the reaction pathways which occur only on the DAD ligand with metal as a spectator. These investigations elucidate the complete mechanism of conversion of **2_m** to **4_m** and water-mediated cleavage of H₂ to regenerate **2_m** from **4_m**. These calculations indicate that conversion of **2_m** to **4_m** is exergonic. However, since the reaction is performed in an open vessel, both species **2_m** and **4_m** should co-exist in solution (although not in a thermodynamic equilibrium), and could act as independent catalysts in the system. This is in line with the experimental observations. While release of H₂ from **4_m** to form **2_m** is certainly possible, please note that the data presented in this paper suggest that this is not the main pathway for H₂ production from methanediol. Dehydrogenation processes occurring on the metal have clearly lower computed barriers than release of H₂ from the ligand.

Below we provide the energy change for hydrogenation of **2_m** to **4_m** at the level of theory we have used in the present work (all values in kcal mol⁻¹, thermochemical values reported at 298 K, correction for change in standard state not included):

Reaction	ΔSCF_{ZPE}	ΔH	ΔG
$2_m + 2H_2 \rightarrow 4_m$	-14.8	-18.1	-0.8
$2_m + 2HCOOH \rightarrow 4_m + 2CO_2$	-30.0	-30.0	-26.4
$2_m + HOCH_2OH \rightarrow 4_m + CO_2$	-27.0	-27.1	-24.8

The small ΔG of -0.8 kcal mol⁻¹ indicates that under an environment of gaseous H₂, **2_m** and **4_m** should be in equilibrium with each other. The fact that the reaction is performed in an open vessel and that hydrogenation of **2_m** by methanediol/formic acid is more exergonic, implies that both **2_m** and **4_m** should be present in significant amounts to carry out the catalysis.

For references see:

Reference 1: H. Li, M. B. Hall, *J. Am. Chem. Soc.* **137**, 12330 (2015). DOI: 10.1021/jacs.5b07444

Reference 2: Y. Jing, X. Chen, X. Yang, *Journal of Organometallic Chemistry* **820**, 55 – 61 (2016). doi: 10.1016/j.jorganchem.2016.07.020.

Conversion of 2_m to 4_m: Figure 5 and Figure 7 in Reference 1. Figure 2 in Reference 2

Conversion of 4_m to 2_m: See Figure 12 in Reference 1. The reported barriers for H₂ generation in this manner are ~35 kcal/mol.

Reviewer 1, comment 2:

In Figure 5, intermediate F is the resting state with the lowest relative free energy of 19.0 kcal/mol, while transition state TS-1 has the highest relative free energy of 15.4 kcal/mol. Therefore, the total free energy barrier of this catalytic reaction is 34.8 kcal/mol, which seems too high for a reaction at 60 °C.

Response:

We thank the reviewer for pointing out the confusing presentation of our data in **Figure 5** (main text). However, species **F** is *not* the resting state. The barriers for **TS-1**, **TS-2**, **TS-3** and **TS-4** should not be calculated from the energy of species **F** for two reasons: (1) The overall catalytic cycle is driven forward by release of H₂ and CO₂ from the system by performing the catalysis in an open reaction vessel. This drives all reaction steps involving gas formation to completion. (2) The regeneration of catalyst **2_m** by release of hydrogen gas from complex **G** is actually an exergonic process. The regenerated complex **2_m** is about 4.6 kcal/mol more stable with respect to **F** and is exergonic by -24 kcal/mol with respect to starting materials (see **Figure S24** below). This information was lacking in the original **Figure 5**, but is added to the corrected new figure. We have also provided a reaction energy profile shown in Figure S24 below to clarify these points. See also response to comment 4.

Reviewer 1, comment 3:

In Figure 5, the reaction starts with methanediol as the reactant. What is the relative free energy for the formation of methanediol from formaldehyde and water? This might be an important step in the catalytic conversion of whole formaldehyde and water to H₂.

Response:

The hydration of formaldehyde in water can be described by equations (1) and (2) (see reference below):

The experimental equilibrium constants (in terms of mole fractions) for equations (1) and (2) are 1300 and 5, respectively, at 298 K (see reference below). The experimental activation energies for equations (1) and (2) in the temperature range of 293 – 333 K are 5.7 kcal/mol and 13.3 kcal/mol, respectively. Hence, formation of methanediol from formaldehyde is a low barrier and energetically downhill process.

Modelling the formation of methanediol in aqueous solution computationally requires explicit solvation by multiple water molecules, which is beyond the scope of this study (see also reference below). Furthermore, addition of base (KOH) will influence the equilibrium further and additionally complicate a theoretical study. Under catalytic conditions, both formalin and paraformaldehyde give a clear solution before the catalyst is added and we assume that methanediol is a major component.

Reference:

Mugnai, M., Cardini, G., Schettino, V. and Nielsen, C.J. *Molecular Physics*, Volume 105, Issue 17-18, **2007**, 2203 – 2210 (<http://dx.doi.org/10.1080/00268970701513864>)

Reviewer 1, comment 4:

In addition to list the relative energies in Figure 5, I suggest authors to provide a free energy profile with reaction coordinate and relative energies for a clearer description of the reaction mechanism.

We thank the reviewer for pointing this out. We agree that providing reaction energy profiles makes the computational data clearer to the reader and helps in the presentation. We have made the necessary changes to the supplementary information file and highlighted the changes. We have modified Figure 5 in main text providing an energy profile diagram. We have also included the new Figures S24 and S25 in the experimental section (shown below).

Figure S24. Methanediol dehydrogenation by catalyst complex 2_m.

Figure S25. Methanediol dehydrogenation by catalyst complex 4_m . (a) Reaction energy profile showing generation of complex $4-F$ and $4-F'$ from methanediol. The pathway marked with red

lines shows the high barrier pathway for formate oxidation via a classical beta-H elimination. The pathway marked with green lines indicates an alternative pathway after flipping a proton on complex **4-D**. (b) Since the reactions are driven forward by escape of gaseous H₂ and CO₂ from the reaction mixture, the barriers for the final dehydrogenation steps leading to formation of **4'm** and **4'm-cis** via **4-TS4** and **4-TS4'** must be computed from species **4-F** rather than **4-D**. Hence these steps are visualized separately. The values in parenthesis are energy values with respect to the starting materials.

Reviewer 1, comment 5:

Please specify clearly which kind of energy, electronic energy, enthalpy, or free energy, is the energies along with the atomic coordinates.

Response:

All the energy values reported along-with the coordinates are in atomic-units. We have made the necessary changes in the SI and highlighted the changes.

Reviewer 1, comment 6:

The names of authors of Ref. 36 seem redundant.

Response:

This reference is correctly cited and all names of the authors are correctly spelled.

Reviewer 2, comment 1:

- *The introduction starts with water as oxygen source for oxygen-transfer reactions. This has been discussed and analyzed by isotope-labelling experiments with labelled water during the alcohol dehydrogenation via aldehydes in detail in Nat. Chem., 5, 122–125,(2013) doi:10.1038/nchem.1536). In 2014, another group observed and described this oxygen transfer reaction in aqueous formaldehyde solution again using oxygen-labelled molecules (Nat. Commun. 2014, 5, Article number: 3621, doi:10.1038/ncomms4621). So this, aspect is not really new. Indeed, while reading the manuscript, I found you related the incorporation of oxygen from water into the well-known carbonate formation. I do not know the original article, but a patent filed in the late 18th century (by Schweppes) described the saturation of water with CO₂, forming aqueous carbonic acid (H₂CO₃), respectively carbonate under basic conditions. In this regards, I feel you over-sell the non-catalytic phenomena of oxygen-transfer from water to CO₂ (yielding sparkling water), the CO₂ is formed during the formaldehyde decomposition, and than redissolved in the water giving carbonate at the applied basic conditions (1-3M KOH for example!). I do not really understood, why have you used the oxygen-transfer from water to carbonate in alkaline media as 'cliff-hanger' in the title of this fine work.*

Response:

We have used the term “Water as Oxygen Transfer Reagent” because this describes best the transformation of formaldehyde to formic acid and hydrogen according to: $\text{H}_2\text{CO} + \text{H}_2\text{O}^* \rightarrow \text{HCOO}^*\text{H} + \text{H}_2$ (“O*” denotes the oxygen atom which is transferred). Formic acid is then decomposed to CO₂ and a further equivalent of H₂. The CO₂ molecule is subsequently sequestered as carbonate under alkaline conditions. As the reviewer points out correctly we made a silly mistake and indeed any hydration reaction like $\text{CO}_2 + \text{H}_2\text{O} \rightarrow \text{H}_2\text{CO}_3$ can be viewed as an “oxygen transfer reaction” as well. However, what we meant and want to point out is that water serves as oxygen transfer reagent in an oxygenation reaction (H₂CO to HCOOH) which proceeds under evolution of hydrogen (that is, this is a formal oxidation reaction - using the language of organic chemistry - which proceeds under “reducing conditions”). It was never our intention to oversell this observation, which we never state to be new nor original. However, not many catalytic processes use water as a source of oxygen and this is described only a few times in the papers cited in our manuscript. We do not insist to keep this term in the title (although we believe it summarizes the findings in our paper properly). We propose the following new title: **“Homogeneously Catalysed Conversion of Aqueous Formaldehyde to H₂ and Carbonate.”**

Reviewer 2, comment 2:

- *In your manuscript (introduction and results) and the description of the state of the art about methanol and formaldehyde dehydrogenation, I recognised you mix two different information in an inappropriate way:*

Adding a base to a system does not imply directly that the reaction has been conducted under basic conditions and the base concentration plays an important role in the system in regards to the pH value. For example the system described for methanol dehydrogenation (Nature 495, 85–89 (2013); Figure 1a, catalyst A), is conducted in a MeOH/H₂O mixture with 8M KOH, unquestionable this is strongly basic (pH = 14); even a solution of 1M KOH/water gives a pH of 14. In contrast in Figure 1b, applying catalyst C, the cited reference, describes the addition of base to "buffer" the in situ formed formic acid and HCl, the reported pH is below 7. Thus, these are no basic conditions at all.

Response:

The reviewer makes a good point here and our formulation is incorrect with respect to the use of catalyst **C**. Hence we reformulate the corresponding passage on page 3 to: "To date, complex **C** is the only catalyst which promotes the dehydrogenation of aqueous formaldehyde solutions to CO₂ and H₂ with acceptable TON and turn over frequencies (TOF) at 95 °C in a phosphate buffered solution at pH = 5.5. With more diluted aqueous formaldehyde solutions (1.6 M) TONs of 700 and TOFs of 3142 h⁻¹ could be achieved even without additives such that this catalytic system can be used for the decontamination of water.[30]"

In reference [29] we write: "For the dismutation of formaldehyde to mixtures of MeOH and formic acid in a buffered solution at pH= 5.5 using catalyst **C** and analogs thereof in a sealed system, see: Waals, D., Heim, L. E., Vallazza, S., Gedig, C., Deska, J. & Pechtl, M. H. G. Self-Sufficient Formaldehyde-to-Methanol Conversion by Organometallic Formaldehyde Dismutase Mimic. Chem. Eur. J. 22, 1 - 7 (2016).

In our catalytic reaction, we use 6 eq. of KOH and it is safe to say that our catalyst works best under "basic conditions".

Reviewer 2, comment 3:

- You probably overlooked one article about formaldehyde dehydrogenation, because you wrote on page 3, that the best catalyst gives a TOF of 170 per hour and a TON of 188. A fast check on Web of Science showed that there is article of 2016 reporting a TOF of 3142 per hour and a TON of 700 under near neutral conditions (Green Chem. 2016, 18, 1469-1474) which is almost 20-times higher than the cited one.

Response:

Thank you for pointing out this other fine communication from Pechtl and co-workers. This communication refers to the decontamination of water polluted with a small weight % of formaldehyde. In this process, the authors use the same catalyst as in reference 28 and for the sake of keeping our manuscript as short as possible, we skipped this paper. However, we agree that this is not justified also because a valuable alternative protocol for the synthesis of arene ruthenium (II) complexes is reported. We included this work now as ref. 30 in the current version of the manuscript (see response to comment 2).

Reviewer 2, comment 4:

Now I must point out the major problems of this study which have been not considered at all:

On page 12 the author state correctly the "... kinetics are strongly dependent on the temperature and pH of the solution." Now I have mention again that the base concentration is very important in this particular case. From organic chemistry textbooks we learn that aldehydes undergo disproportionation to the corresponding alcohols and carboxylic acids under basic conditions (pH >11). This reaction is known as the Cannizzaro reaction (https://en.wikipedia.org/wiki/Cannizzaro_reaction; S. Cannizzaro, 1853: doi:10.1002/jlac.18530880114). The authors applied an excess of 2, 4, 6 eq. of KOH to aq. formaldehyde in most reactions (refer ESI and discussion). The given concentrations are equal to 1-3M solution of KOH in aq. formaldehyde. Undiscussable these are strongly alkaline conditions with a pH 14. Heating aqueous formaldehyde to to 60°C at pH 14 results in the formation of methanol and formic acid. This could be easily checked with a blank experiment and analysis by 1H-NMR to confirm the well known Cannizzaro reaction (it would be very surprising if the authors could reveal that the Cannizzaro reaction do not take place in basic media!). Luckily, the authors applied a catalyst which is known to be active also for methanol dehydrogenation (Nat. Chem. 2013, 5, 342–347.) - more about this aspect on a later point. Considering the Cannizzaro reaction and the obvious formation of methanol and formic acid, the catalyst dehydrogenates in the first instance the formic acid to CO₂ and H₂ with a TON of >1700 (TOF of 12000 per hour) which is not so fast and there are many publications on formic acid decomposition giving much higher TONs in the range of >100.000. Interestingly the authors report on page 12 that "Full conversion, that is the release of two equivalents of hydrogen, was never achieved". One equivalent yields from the formic acid decomposition, and some amounts are generated from the methanol dehydrogenation which is probably slower, more over the authors reported residual potassium formate which is not fully decomposed. The H₂ derived from methanol can be estimated by NMR analysis. Most interestingly, the catalyst 1Aa (page 14) decomposed paraformaldehyde with a TOF of >20.000 per hour, yielding 90% of H₂. This is very good.

- However the proposed reaction pathway based on DFT calculations (Figure 5) is probably more complex. 1. Cannizzaro reaction occurs, 2. formic acid is dehydrogenated to H₂ and CO₂ (respectivel carbonate under basic conditions), 3. methanol is dehydrogenated to formaldehyde and H₂, 4. the generated formaldehyde from the methanol dehydrogenation undergoes in the next cycle a Cannizzaro reaction again, and so on.... So, the Cannizzaro reaction, in this particular case, is not a dramatic story, because a catalyst has been used which is luckily active for the methanol dehydrogenation. Therefore, the reaction pathways are simply different and a new proposed mechanism is required to be included into the publication - an extended cycle

Response:

The reviewer makes again a very good point here. Indeed, it is well known that formaldehyde disproportionates to formic acid and methanol under strong basic conditions and heating. A considerable amount of work has been devoted on studying the kinetics of this reaction (see for example a study under supercritical conditions by A. Kruse and co-workers, Appl. Catal., A 2003, 245, 333, added as reference 43). We actually considered this reaction but did not sufficiently refer to the results in our manuscript. We apologize for this. We have now modified the main text accordingly and also performed additional experiments which make it very unlikely that the Cannizzaro reaction is the main pathway for hydrogen evolution. Note, however, that we cannot and do not exclude that this reaction occurs as side-reaction. As we describe now in the text, a blank experiment without catalyst gives very low conversion in the reaction $2 \text{H}_2\text{C}=\text{O} + \text{H}_2\text{O} \rightarrow \text{HCOOH} + \text{MeOH}$. When a 1:1 mixture of HCOOH and MeOH are reacted in presence of the catalysts under basic conditions, hydrogen is formed but again with much lower efficiency. MeOH is not notably converted under these conditions. Note also that we do not detect accumulation of MeOH when aqueous formaldehyde solutions are catalytically converted to H_2 and carbonate. These results show that the Cannizzaro reaction cannot be the major pathway on which hydrogen is produced. We have therefore not further investigated this reaction. Admittedly, it remains somewhat unsatisfying that we do not understand why the conversions of aqueous formaldehyde do not proceed to completion. We always observe residues of formate (ca. 5%). We also observe that our catalysts do convert solutions of formate, $\text{M}(\text{O}_2\text{CH})$ with M = alkaline ion or ammonium – much slower than formic acid itself. Why this is the case remains speculative (some kind of product poisoning?) and must be further investigated eventually in collaboration with other groups.

Specifically we changed the text of the manuscript as follows:

a) page 12, introduction to catalysis part: “The dehydrogenation of aqueous formaldehyde solutions can proceed following different pathways: i) The Cannizzaro reaction, simplified as $2 \text{H}_2\text{C}=\text{O} + \text{H}_2\text{O} \rightarrow \text{HCOOH} + \text{H}_3\text{COH}$, [43= A. Kruse and co-workers, Appl. Catal., A 2003, 245, 333] followed by catalytic dehydrogenation of methanol and formic acid as previously reported.[9,15]. ii) The decarbonylation of formaldehyde according to $\text{H}_2\text{CO} \rightarrow \text{H}_2 + \text{CO}$ followed by a water-gas shift reaction, $\text{CO} + \text{H}_2\text{O} \rightarrow \text{CO}_2 + \text{H}_2$. iii) Direct dehydrogenation of methanediol according to $\text{H}_2\text{C}(\text{OH})_2 \rightarrow \text{HCOOH} + \text{H}_2$ followed by decomposition of formic acid, $\text{HCOOH} \rightarrow \text{CO}_2 + \text{H}_2$.”

b) page 15, final section of the catalysis part: “We considered the possibility that formaldehyde disproportionates in a Cannizzaro reaction as shown in (i). In that case, the complexes listed in Table 2 may merely catalyse the dehydrogenation of formic acid and methanol. In a blank reaction without any Ru catalyst, formaldehyde was heated with 6 equivalents of KOH in D_2O at 60 °C to give only 6% conversion after 15 minutes. The conversion increases to about 30% HCOOH/MeOH after 12 h. In a second experiment, a 1:1 mixture of MeOH and formic acid was heated to 60 °C in the presence of complex **1Aa** under conditions given in entry 5 of Table 2. In this experiment, 27% of H_2 is evolved. This corresponds to the expected amount of H_2 from the

decomposition of formic acid (25%). Analysis of the reaction mixture by NMR reveals the presence of 4% unconverted formate and 97% of methanol. Hence, methanol is not converted under these conditions.”

c) page 15, introduction to the DFT study: “Likewise, the possible disproportion of formaldehyde to formic acid and methanol was not further investigated. This reaction may be a side-reaction but is unlikely to be the major reaction pathway (vide supra).”

Reviewer 2, comment 5:

- A positive aspect is that the reaction can be run under CO atmosphere (page 15) which opens new possibilities. On the same page they mention the major drawback of their system, besides the strong basic conditions (!; 1-3 M KOH in aq. formaldehyde!), the catalyst is just stable in deoxygenated solvents and in absence of air! Under air and non-deoxygenated water, the catalyst is completely inactive and cannot compete with air-stable catalysts.

Response:

This is a drawback of the catalyst, indeed, as we state in the paper. We are aware of the fact that from these results to a real working device or an industrial process there is a very long way to go. However, we believe that oxygen intolerance is not the major problem because any device producing larger amounts of hydrogen will not be operated in the presence of oxygen. Note that most of the complexes reported here are stable, can be stored for longer periods of time, and handled briefly on air. We believe that an important observation is that a) some new insights into possible mechanisms is gained and b) that the problem of CO contamination may be solved, which in the case of an aldehyde (and especially formaldehyde) is a high risk.

Reviewer 2, comment 6:

- On page 18, the authors state again that carbonate is formed under basic conditions. I would say that this is again a drawback in regards to the accumulation of a salt in the liquid phase, contrary one could argue the purity of the delivered H₂ is higher. However the carbonate must be recycled or separated in some way.

Response:

While this can be considered as a drawback, it is in our opinion still better than the production of gaseous CO₂. The carbonate trap produces a pure H₂ stream. While the presence of CO₂ can lead to the formation of CO inside in the fuel cell on the Pt surface by hydrogenation of CO₂ via the reverse water gas shift (RWGS) reaction or the electrochemical RWGS (see for example: T. R. Ralph et al. Platinum Met. Rev. 2002, 46, 117-135). Small amounts of COx (traces in the case of CO) in the fuel cell feed-stream causes a dramatic decrease in performance, especially at high current densities and low temperatures (see for example: A. Kaufman et al. Electrochem.

Solid-State Lett. 2001, 4(12), A204-A205). Carbonate – as a concentrated form of CO₂ – can in principle easily be recycled to hydrogenated products.

Reviewer 2, comment 7:

- On page 18, the author state again: "It is very likely that water serves as oxygen transfer reagent and in reactions of organic aldehydes with O18 labelled water, RCH=O16 + H2O18 → RCO18/16O18H + H2, this hypothesis was proven.45" Ref. 45 states: "45 In the homogenously catalysed reaction of benzaldehyde with H2O18 with complexes 1 and 3a PhCO18O18K/ PhCO16O18K/ were obtained. These results will be published separately. See also reference 15." I would say that this has already been proven by means of isotope-labelling experiments that water serves as oxygen transfer reagent, vide supra and refer: Nat. Chem., 5, 122–125,(2013) and Nat. Commun. 2014, 5, Article number: 3621, doi:10.1038/ncomms4621. Please do not use the oxygen transfer from water to aldehyde and water to CO2 as 'cliff-hanger'. This is not really necessary and already known.

Response:

As we stated above, it is not at all our intention to use the simple fact that water serves as oxygen transfer reagent as “cliff-hanger”. It is in our opinion just a very good description of what actually happens in these reactions. Please note, that these types of reactions are sometimes denoted as “oxidation” reactions, which in our opinion is a rather incorrect description with respect to the mechanism. An oxidation reaction requires the removal of electrons which is highly unlikely with electron rich metal complexes such as Ru(0) compounds used here. For that reason, we kindly ask to leave these formulations in the text. We modified reference 45 to: “In the homogenously catalysed reaction of benzaldehyde with H₂O¹⁸ with complexes **1** and **3a** PhCO¹⁸O¹⁸K/ PhCO¹⁶O¹⁸K/ were obtained. These results will be published separately. See also references 15 and 28.” That is we do refer now to references 15 and 28.

Reviewer 2, comment 8:

In summary, the paper gives some interesting insights about new catalysts for the H2 generation from formaldehyde under basic conditions in oxygen-free solvent and atmosphere. The report needs a revision to become suitable to be published in this journal - see comments above. The major concerns are the basic conditions which lead to the well-known Cannizzaro reaction pathway. Consequently, the whole reaction pathway is different, but luckily owing to the use of the right catalyst which is capable for methanol dehydrogenation high yields of H2 are still possible. The authors reached their target, but via an overlooked and unconsidered but well-known reaction pathway.

Response:

As outlined above, we actually considered the possibility that formaldehyde is converted to formic acid and methanol via the Cannizzaro reaction. However, while this reaction may be a

side reaction it is not the main reaction pathway. We refer to this by stating in the paper: “The experimental stoichiometric reactions and catalytic studies clearly indicate that there are several ruthenium species formed under catalytically relevant reaction conditions, and likely several mechanisms are operative in the conversion of aqueous formaldehyde solutions to carbonate and hydrogen under basic conditions.” We actually believe that the value of our work relies exactly in the fact that metal complexes were found which are likely able to work along different reaction pathways and are not easily deactivated. Furthermore we propose a mechanism which has not been reported in this form previously and in our opinion gives a better picture of a pathway which does not involve unrealistically high activation barriers. But as many ways lead to Rome this city was also not build in one day. Clearly much – especially collaborative - work remains to be done before renewable resources can be efficiently used as energy carriers.

Reviewer 3, comment 1:

There are a number of issues with the crystallographic data in the body of the paper as well as in Supplementary Material. In order to perform a thorough review, I attempted to run the 6 CIF files through the standard checkCIF review. Most of the CIFs contained syntax errors which prevented a checkCIF analysis until I made the necessary edits. This should have been done by the authors before submission. In addition, one of the CIFs did not include hkl data which prevented a complete analysis.

Response:

We deeply apologize for these inadvertencies. All syntax errors have been corrected. The missing hkl data for complex **7** has been appended in the corresponding cif file.

Reviewer 3, comment 2:

The data for 1Aa were included in the SI but little detail was provided in the text as to its formulation. The CIF contains 19! partial occupancy Br sites as well as a partial occupancy hexane site including hydrogens. Clearly, this was just an attempt to "mop up" residual electron density in the Fourier map but anything could have been modeled at these sites. Therefore, the formulation for 1Aa has not been unambiguously established from x-ray data...nor was there any attempt to provide it in the CIF. At the very least, a comment should have been included in the CIF with respect to the choice of model.

Response:

Structure **1Aa** has been solved and refined now in a different way. The disorder of the bromine anion was modelled over 2 main sites, one of which was removed via mask. Site one contains approximately 75 % of the bromines electron density, while the masked site 2 contains the remaining density and is in addition overlapped with half a molecule of n-hexane. The mask was applied because previous attempts at modelling site 2 did not result in a reasonable / stable structure refinement. The masked electron density (41.4 e per cell) was in accordance with the model and charge balance for the ammonium salt is achieved. A comment is included in the cif file.

Disorders of Bromine anions in ammonium salts are not uncommon and can be readily found in the CCDC (e.g. CCDC 643823). Updated metric data is provided in the supplementary info and Table 1 in the main text.

Reviewer 3, comment 3:

Several of the 6 CIF files generated A and B level alerts in checkCIF. Multiple other alerts were generated by checkCIF but these can generally be read and ignored. However, comments regarding the A and B alerts should be included in the CIFs.

Response:

All the structures have been revised and no alerts A are reported. Comments regarding the level B and C alerts were added to the corresponding cif files under `_refine_special_details`.

Reviewer 3, comment 4:

For compound 1Ab, the Ru-H distance is unrealistically short (1.19Angstrom)(based on a survey of the Cambridge Database). While the location and refinement of hydrides in the presence of heavy atoms is a well-known crystallographic problem, at the very least, a comment recognizing this anomaly is required.

Response:

A comment regarding the hydrogen (metal hydride) position in the structure was added to the cif file.

Reviewer 3, comment 5:

The Figure captions for Figure 4 contain several errors as well. It's apparent that the final names in the CIFs do not match those in the Figure captions and at some point, the structures were renamed. As such, it makes it quite difficult to follow the authors discussion. For example, the values given in caption 4d for C4-C5 and C19-C20 do not match the values in the CIF file. For Figure 4b, Ru-N and other Ru distances are reported in the caption but no explanation for Ru2-ct5? and no discussion in the text? In addition, Ru - N and Ru-C distances aren't provided for any of the other structures so why include them here? In the discussion, standard uncertainties are not provided on several of the reported values making it difficult to assess the magnitude and significance of the differences.

Response:

We apologize for these mistakes, an old version of the cif file of structure **1Aa** was provided. The bond lengths in the figure caption for Figure 4 have been removed and can now be found in Table 1, which was transferred from the SI (formal Table S2) to the main text and completed with additional data (Ru-ct5 bond distances). Now all the atom labels should match the ones in the CIF files. The errors were corrected and missing bond lengths and standard uncertainties were added to the table and text. In the SI, the captions for figures S14-S19 and Table S3 were updated / corrected. Brief discussion of Ru-N bonds is included in the main text.

Reviewer 3, comment 6:

Clearly, the authors have done a great deal of work in the preparation of this manuscript. I sympathize that the preparation of papers reporting multiple structures is tedious. However, the work here is incomplete. The crystallographic data could do a great deal to support the authors

arguments but it needs to be "cleaned up" and better presented. At the very minimum, accurate formulas should be given in the text for each of the structures (including solvent/disorder); i.e. what is 1Aa vs. 1Ab, etc. For the discussion of the metrical structural parameters, it would be much better for the values to be compared in tabular form as opposed to figure captions; i.e. Table S2 would be a valuable addition to the text as opposed to buried in supplementary material. Or, at least, the existence of the table should be referenced in the text. There's a great deal of supportive evidence here but the authors have hidden it and could make it much easier to find.

After significant revision and additional refinement for some of the structures with corrected CIFs, I believe that this paper should be reconsidered for publication.

Response:

All the structures include now the complete formulas in the main text. The Table 1 is now present in the text and contains all relevant bond lengths.

Reviewers' comments:

Reviewer #1 (Remarks to the Author):

This revised manuscript has addressed most of my concerns. I would like to support its publication with the following minor revisions.

1. In the energy profile diagram shown in Figure 5, the highest barrier is only 17.1 kcal/mol (E → TS-4), which is too low for a reaction at a temperature of 60 °C. I was wondering if there is any stable intermediate missed in this energy profile. A brief discussion for this inconsistency would be very helpful.
2. Some 3D structures of key transition states would be helpful in understanding the reaction mechanism.
3. In addition to the SCF energy, please also provide the absolute free energies of all calculated structures along with their atomic coordinates in the supporting information.
4. The names of authors appear twice in Ref. 37.

Reviewer #2 (Remarks to the Author):

Dear authors,

I have gone through your revised submission and rebuttal letter. You tried to comment on all questions, but you raised new doubts with some modifications and comments. Please find my comments on your answers below:

Reviewer 2, comment 1:

Response:

"[...]We have used the term "Water as Oxygen Transfer Reagent" because this describes best the transformation of formaldehyde to formic acid and hydrogen according to: $\text{H}_2\text{CO} + \text{H}_2\text{O}^* \rightarrow \text{HCOO}^*\text{H} + \text{H}_2$ ("O*" denotes the oxygen atom which is transferred).[...]"

Reviewer comment:

Indeed, this has nothing to do with your catalyst system and is not completely true. The "oxygen transfer reaction" takes place previous to the dehydrogenation of methanediol to formic acid. The oxygen is already transferred by mixing water with formaldehyde readily forming methanediol (equilibrium constant: $K = \sim 2000$)! In other words, using commercially available aqueous formaldehyde solutions, you already bought methanediol and the oxygen transfer reaction took place even before the batch arrived in your lab! This observation of methanediol formation has been reported several decades ago, therefore I would modify this part previously to publication.

Reviewer 2, comment 4:

Response:

"[...] We have now modified the main text accordingly and also performed additional experiments which make it very unlikely that the Cannizzaro reaction is the main pathway for hydrogen evolution. Note, however, that we cannot and do not exclude that this reaction occurs as side-reaction. As we describe now in the text, a blank experiment without catalyst gives very low conversion in the reaction $2 \text{H}_2\text{C}=\text{O} + \text{H}_2\text{O} \rightarrow \text{HCOOH} + \text{MeOH}$.

[...] These results show that the Cannizzaro reaction cannot be the major pathway on which hydrogen is produced. We have therefore not further investigated this reaction. Admittedly, it remains somewhat unsatisfying that we do not understand why the conversions of aqueous formaldehyde do not proceed to completion. We always observe residues of formate (ca. 5%).

[...] The conversion increases to about 30% HCOOH/MeOH after 12 h. In a second experiment, a 1:1 mixture of MeOH and formic acid was heated to 60 °C in the presence of complex 1Aa under conditions given in entry 5 of Table 2. In this experiment, 27% of H₂ is evolved. This corresponds to the expected amount of H₂ from the decomposition of formic acid (25%). Analysis of the reaction mixture by NMR reveals the presence of 4% unconverted formate and 97% of methanol. Hence, methanol is not converted under these conditions."

Reviewer comment:

With your statement: "a blank experiment without catalyst gives very low conversion in the reaction $2 \text{H}_2\text{C}=\text{O} + \text{H}_2\text{O} \rightarrow \text{HCOOH} + \text{MeOH}$." you state that you cannot reproduce the century old Cannizzaro reaction?! Under basic conditions, $\text{pH} > 11$, formaldehyde disproportionates to formic acid and methanol. Your conditions are $\text{pH} 14$ with 3M KOH solutions. And I really wonder how you suppress the Cannizzaro reaction?! This leaves me puzzled and I must say that to my experimental knowledge heating aqueous formaldehyde at $\text{pH} 14$ the Cannizzaro reaction in much shorter reaction time ($\ll 12\text{h}$) yields MeOH and formic acid.

The statement: "Analysis of the reaction mixture by NMR reveals the presence of 4% unconverted formate and 97% of methanol." leaves me puzzled again. The authors 2013th Nature Chemistry paper about methanol dehydrogenation give high yields on methanol dehydrogenation. Which parameter suppresses now the methanol decomposition in comparison to your original work?

Moreover, it should be noted that the hydroxide ions from KOH are present in much higher concentrations than the ruthenium catalyst. Thus it is statistically (much lower probability) completely unlikely that methanediol is dehydrogenated to formic acid by the ruthenium complex previously to a reaction before methanediol reacts with hydroxide ions at $\text{pH} 14$ (3M KOH). Any explanation why this should occur in your system? Therefore, I still think that the proposed/calculated reaction pathway is idealised and it does not take into account the strong alkaline conditions and the Cannizzaro reaction.

In comparison to other original comments, this point remains to be clarified previous to publication in this journal.

Reviewer 2, comment 7:

Response:

"[...] It is in our opinion just a very good description of what actually happens in these reactions. Please note, that these types of reactions are sometimes denoted as "oxidation" reactions, which in our opinion is a rather incorrect description with respect to the mechanism.[...]"

Reviewer comment:

As stated above, the oxygen transfer reaction occurs previous to the dehydrogenation and also in absence of your catalyst! And it is true, it is a hydration of an aldehyde and not the oxidation of an aldehyde to a carboxylic acid.

After clarifying the above described new doubts, I clearly support the publication of this manuscript in a revised form in this journal.

Reviewer #3 (Remarks to the Author):

I appreciate that the authors have addressed my comments/concerns regarding the crystallographic details in the original manuscript and associated materials. I am, therefore, satisfied that the manuscript has been adequately revised to include these changes and, as such, I support publication in the current revised form.

We would like to thank again the referees of this manuscript for the helpful comments and remarks, which certainly have improved the manuscript. Our responses to the referees #1 and #2 are indicated below. All the new changes in the second revision have been highlighted in yellow in the main text and experimental section. Previous changes from the first revision are still marked in grey.

Reviewer 1: Comment 1

In the energy profile diagram shown in Figure 5, the highest barrier is only 17.1 kcal/mol (E → TS-4), which is too low for a reaction at a temperature of 60 °C. I was wondering if there is any stable intermediate missed in this energy profile. A brief discussion for this inconsistency would be very helpful.

Response:

We have performed an exhaustive search to try that every relevant intermediate in the catalytic cycle has been accounted for. There is no full-proof method to ensure that no stable intermediate has been missed in DFT studies, but we do not think we missed an obvious one. Perhaps explicit solvation effects stabilize species **E** stronger than the **TS**, which would increase the energy barrier. Additionally, as noted by Li and Hall, the simplified model we used in the calculations (truncated trop ligand) sometimes underestimates the transition state barriers as compared to the catalyst with the full-atom ligand for steric reasons. However, addressing all these issues is beyond the scope of the present paper. In the present paper we primarily want to show that the metal-mediated processes proceed over pathways with much lower barriers than the reactions in which only ligand centered reactivity is assumed as reported previously (Hall and Li). The goal of the computational studies in our paper is not to provide a quantitative reproduction of the experimental barriers (which in fact are not known in detail to date anyway). The study rather focusses on trends and aims at providing a qualitative picture of the most likely pathways.

In the main text we added the yellow marked sentences in the DFT section to explain these issues (page 18):

“This process has the highest barrier (+17 kcal mol⁻¹) in the catalytic cycle, and seems to be the TOF-limiting step for catalyst **2_m**. The computed overall barrier for the reaction seems to be somewhat low for a reaction requiring heating in the experimental reactions. The apparently underestimated barrier might be due to the simplified ligand used in the computational studies (truncated trop moiety), unaccounted explicit solvation effects in the gas phase DFT calculations, and/or limitations of the functional used. However, addressing all these issues is beyond the scope of the present paper, which aims at providing a qualitative picture of the most likely pathways occurring at ruthenium. The resulting formation of **F'** is quite exergonic, and loss of CO₂ from **F'** to form **F** is further downhill on the energy landscape.”

Reviewer 1: Comment 2

Some 3D structures of key transition states would be helpful in understanding the reaction mechanism.

Response:

We thank the reviewer for this suggestion. We have added in the ESI Figures S26 and S27 to show 3D structures of all the transition states involved in the mechanism. See also below:

Figure S26. Graphical representation of transition states **TS-1** to **TS-5** (dashed lines indicate relevant bond lengths (Å) in the transition state).

Figure S27. Graphical representation of transition states **4-TS1** to **4-TS4'** (dotted lines indicate H-bonding, thicker dashed lines indicate relevant bond lengths (Å) in the transition state).

Reviewer 1: Comment 3

In addition to the SCF energy, please also provide the absolute free energies of all calculated structures along with their atomic coordinates in the supporting information.

Response:

We had already presented the SCF energies, ZPE corrected SCF energies, enthalpy and free energy data under the section heading ENERGY TABLES just before the references in the ESI.

For clarity, we decided to move the energy tables now to a position in the supporting information BEFORE the coordinates.

Reviewer 1: Comment 4

The names of authors appear twice in Ref. 37.

Response:

The author names have now been corrected and we thank the reviewer for pointing this out to us.

Reviewer 2, comment 1:

> Response:

> "[...]We have used the term "Water as Oxygen Transfer Reagent" because this describes best the transformation of formaldehyde to formic acid and hydrogen according to: $H_2CO + H_2O^* \rightarrow HCOO^*H + H_2$ ("O*" denotes the oxygen atom which is transferred).[...]"

>

> Reviewer comment:

> Indeed, this has nothing to do with your catalyst system and is not completely true. The "oxygen transfer reaction" takes place previous to the dehydrogenation of methanediol to formic acid. The oxygen is already transferred by mixing water with formaldehyde readily forming methanediol (equilibrium constant: $K = \sim 2000$)! In other words, using commercially available aqueous formaldehyde solutions, you already bought methanediol and the oxygen transfer reaction took place even before the batch arrived in your lab! This observation of methanediol formation has been reported several decades ago, therefore I would modify this part previously to publication.

Response:

We admit that we do not fully understand the concerns of the reviewer and his reluctance to accept that water acts as oxygen transfer reagent. From our point of view this is just a formalism which describes the conversion of formaldehyde to carbonate. This reaction can certainly be also performed using gaseous formaldehyde which is bubbled into an aqueous reaction medium containing the catalyst. The reaction equations would read: $H_2C=O + H_2O \rightarrow H_2C(OH)_2$; $H_2C(OH)_2 \xrightarrow{\text{cat}} HCOOH + H_2$; $HCOOH \xrightarrow{\text{cat}} CO_2 + H_2$. Would in this case the wording be acceptable? We believe so. Although we did not perform that experiment, it is very likely that exactly the same chemistry will occur as if an aqueous solution of formaldehyde is used. Once again, we would like to make clear that it is far from our intention to use the phrase "water as oxygen transfer reagent" as "selling argument" – also because we did not invent this term. We just would like to distinguish between an oxygenation of an aldehyde using water from other oxygenation reagents. We fully agree with the referee that formaldehyde is hydrated first and then dehydrogenated. Yet formally the oxygen atom comes from water. Actually, likely in many oxygen transfer reactions, there will be an intermediate (like methanediol) on the way to the final product. But commonly one would refer to the starting materials when labelling the reaction. In any case, we have removed the wording "water as oxygen transfer reagent" in relation to the reactions which we have investigated and only refer once on page 19 to water as source of oxygen in order to make clear that accidental O_2 is not likely the source of it (see all passages marked in yellow in the text).

Reviewer 2, comment 4:

Response:

"[...] We have now modified the main text accordingly and also performed additional experiments which make it very unlikely that the Cannizzaro reaction is the main pathway for hydrogen evolution. Note, however, that we cannot and do not exclude that this reaction occurs as side-reaction. As we describe now in the text, a blank experiment without catalyst gives very low conversion in the reaction $2 H_2C=O + H_2O \rightarrow HCOOH + MeOH$.

[...] These results show that the Cannizzaro reaction cannot be the major pathway on which hydrogen is produced. We have therefore not further investigated this reaction. Admittedly, it remains somewhat

unsatisfying that we do not understand why the conversions of aqueous formaldehyde do not proceed to completion. We always observe residues of formate (ca. 5%).

[...] The conversion increases to about 30% HCOOH/MeOH after 12 h. In a second experiment, a 1:1 mixture of MeOH and formic acid was heated to 60 °C in the presence of complex 1Aa under conditions given in entry 5 of Table 2. In this experiment, 27% of H₂ is evolved. This corresponds to the expected amount of H₂ from the decomposition of formic acid (25%). Analysis of the reaction mixture by NMR reveals the presence of 4% unconverted formate and 97% of methanol. Hence, methanol is not converted under these conditions."

Reviewer comment:

With your statement: "a blank experiment without catalyst gives very low conversion in the reaction $\text{H}_2\text{C}=\text{O} + \text{H}_2\text{O} \rightarrow \text{HCOOH} + \text{MeOH}$." you state that you cannot reproduce the century old Cannizzaro reaction?! Under basic conditions, pH >11, formaldehyde disproportionates to formic acid and methanol. Your conditions are pH 14 with 3M KOH solutions. And I really wonder how you suppress the Cannizzaro reaction?! This leaves me puzzled and I must say that to my experimental knowledge heating aqueous formaldehyde at pH 14 the Cannizzaro reaction in much shorter reaction time (<< 12h) yields MeOH and formic acid.

The statement: "Analysis of the reaction mixture by NMR reveals the presence of 4% unconverted formate and 97% of methanol." leaves me puzzled again. The authors 2013th Nature Chemistry paper about methanol dehydrogenation give high yields on methanol dehydrogenation. Which parameter suppresses now the methanol decomposition in comparison to your original work?>

Response:

In our statement "a blank experiment without catalyst gives very low conversion in the reaction $\text{H}_2\text{C}=\text{O} + \text{H}_2\text{O} \rightarrow \text{HCOOH} + \text{MeOH}$ " refers to a reaction with a formaldehyde concentration in water identical to the catalytic experiments (0.5 M). We have repeated the reaction under higher concentration (5M, 10-fold) and the Cannizzaro reaction gives indeed a good conversion to formate and MeOH in less than 2h. Under these conditions, the disproportionation of aqueous formaldehyde under alkaline conditions is perfectly reproducible. The Cannizzaro reaction is second order in formaldehyde and requires higher concentrations to occur at a significant rate. However, the relatively low concentration of formaldehyde in our catalytic experiments suppresses the Cannizzaro reaction.

We have performed a number of additional experiments. In one experiment, a 5M solution of formaldehyde is heated at pH 14 for 2 hours to promote first the Cannizzaro reaction and subsequently the catalyst was added and the hydrogen evolution was followed under the conditions as listed in Table 2 (60 °C, 6 eq. KOH, 0.4 mol% cat). Under these conditions, the reaction only achieves 22% of the theoretical H₂ yield and does not proceed further. Analysis of the reaction mixture shows the presence of MeOH, carbonate and traces of formate. Most of the MeOH remains unconverted.

When such concentrated formaldehyde solutions are used in the catalytic reactions, an insoluble red solid forms which we were unable to characterize. But the mass is significantly higher than the initial mass of the metal complex. Although we have no proof for this at the moment, we speculate that under these conditions, the diazadiene moiety is degraded to trop amine species which may form oligo/polymeric materials with formaldehyde. This reaction may lead to catalyst deactivation. By NMR

spectroscopy a very complex mixture of Ru hydrides are observed which we found impossible to characterize any further.

We have also briefly investigated the effect of the concentration of the aqueous formaldehyde (formalin) concentration on the catalytic performance. When the formalin concentration is decreased to 2.5, 1.25 and 0.75 M, the insoluble red solid is not formed. With a 2.5 M concentration of formalin, the catalyst promotes the formation of 32% H₂ yield after 12h. At lower concentrations, higher H₂ yields (73-90%) are obtained and TOFs at 50% conversion increase with decreased formalin concentrations (1.25 M TOF = 1875 h⁻¹; 0.75 M TOF = 7500 h⁻¹) which remain, however, still lower than the one at 0.5 M concentration.

Clearly these phenomena merit a further in depth investigation which is planned and under way. But these results show in our opinion that the Cannizzaro reaction cannot be the major pathway under the conditions reported in the manuscript. Actually, conditions under which the Cannizzaro reaction occurs efficiently must presently be avoided especially because a damage of the catalyst is provoked. It may very well be possible, that a more stable catalyst would tolerate the Cannizzaro reaction as first step and then decompose MeOH and formic acid in the second. We are currently working on such modified catalysts and hope to obtain a complex that catalyzes all possible ways to convert aqueous formaldehyde. This would likely be the best choice for the production of hydrogen from concentrated formalin solutions.

Finally, to answer the question of the reviewer why MeOH is not notably converted with the present catalytic system, we point out that the conditions under which we used catalyst **e** for MeOH/H₂O conversion as reported in Nat. Chem. in 2013 are quite different. First, a significantly lower base concentration was used and secondly the reaction temperature was higher (90 °C) which is necessary to achieve good conversion which still lasted 10 h. Additionally, the methanediol concentration was very low at all times in the MeOH/H₂O conversion and consequently the catalyst was not decomposed rapidly.

In the main text of our manuscript (page 15) we refer now to these observations as follows:

“We considered the possibility that formaldehyde disproportionates in a Cannizzaro reaction as shown in (i). In that case, the complexes listed in Table 2 may merely catalyse the dehydrogenation of formic acid and methanol. When formaldehyde (0.5 M) is heated with 6 equivalents of KOH in D₂O at 60 °C (without any Ru catalyst), which corresponds with the conditions used in the Ru-catalysed reactions, only 6% conversion is obtained after 15 minutes. The conversion increases to about 30% after 12 h. It is known that the Cannizzaro reaction proceeds with high efficiency at higher concentration⁴⁵ and indeed with a 5 M formaldehyde solution at pH = 14 and 60 °C > 90% conversion to formate and MeOH is achieved. When the catalyst **1Aa** is added to this mixture and heated to 60 °C only 22% of H₂ is evolved. This corresponds approximately to the expected amount of H₂ from the decomposition of formic acid (25%). Analysis of the reaction mixture by NMR reveals traces of formate (<2%) and 97% of methanol. Furthermore, a rather rapid decomposition of the Ru complex to an insoluble red solid was observed.

Hence, methanol is not converted under these conditions which is in contrast to our previous report where complex **1K** was found to convert methanol/water mixtures but at much lower base concentrations and higher temperatures.¹⁵ Note also that generally the efficiency of the catalysis decreases with increasing formaldehyde concentration above 0.5 M in water (see Table S2 in the supporting information). The higher catalytic efficiency using diluted formaldehyde solutions was also observed previously by Prechtl *et al.*³⁰

A new reference 45 [Pfeil. E. Über den Mechanismus der Cannizzaroschen Reaktion. *Chem. Ber.* **84**, 229 – 245 (1951)] has been included, regarding kinetic studies of Cannizzaro reaction of formaldehyde.

Reviewer 2, comment:

Moreover, it should be noted that the hydroxide ions from KOH are present in much higher concentrations than the ruthenium catalyst. Thus it is statistically (much lower probability) completely unlikely that methanediol is dehydrogenated to formic acid by the ruthenium complex previously to a reaction before methanediol reacts with hydroxide ions at pH 14 (3M KOH). Any explanation why this should occur in your system? Therefore, I still think that the proposed/calculated reaction pathway is idealised and it does not take into account the strong alkaline conditions and the Cannizzaro reaction.

Response:

We are thankful to the reviewer because his remark made us aware of the important fact that our catalytic system refers to a biphasic system which we did not sufficiently mention in the previous versions. We apologize for this.

The reviewer is right that the concentration of methanediolate is likely higher than methanediol in the aqueous phase. As explained above, this does however not lead to an efficient disproportionation (Cannizzaro reaction) because of the low formaldehyde concentration. But the hydride transfer from methanediolate to the catalyst could still take place.

Before we address this possibility, please note first that the overall reaction for dehydrogenation of aqueous formaldehyde is the following:

From methanediolate alone, two equivalents of H₂ cannot be evolved. An additional proton source is needed, which is (in this case) water. Hydride transfer from methanediolate to ruthenium as a step in the computed cycle is perhaps conceivable. However, this must be followed by protonation of thus formed anionic species to enable H₂ formation in the subsequent steps. As such, only the energies of the first few steps of the computed catalytic cycle (from species of type A to species of type B) are perhaps changed under the assumption that the anionic methanediolate attacks the metal rather than the neutral methanediol. After these steps, the reaction should follow essentially the same pathway(s) as already discussed for neutral methanediol. Hence, there is likely only a small change in MERPs.

There is also an important experimental argument pointing in favor of pathways involving neutral methanediol (as computed). The ruthenium catalyst is added in an organic phase (THF) to the reaction vessel, and the reaction mixture is actually a biphasic system. The water phase contains mainly methanediolate, while the strong color of the organic phase suggests most of the catalyst is present in the organic phase. Hence, most likely the catalytic dehydrogenation reaction takes place in the organic phase. The substrate must thus be transferred from the aqueous phase to organic phase for catalysis to

occur. Anionic methanediolate must therefore convert to neutral methanediol, which is more soluble in the organic phase. As such, we believe that the calculated mechanism as presented describes the experimental catalytic cycle in a consistent manner.

We truly apologize that we did not mention this likely important detail in the main text before. We do this now and on page 12 we write: "THF solutions of complexes **1K**, **1KC**, **1Aa**, **1Ab**, **3a**, and **4 - 7** were tested in the catalytic decomposition of various formaldehyde/water mixtures at 60 °C using a reflux condenser under an inert atmosphere of argon. The progress of the catalytic conversion in this biphasic system was followed by real-time volumetric measurements of released H₂ gas."